# Driving Forces of Changing Environmental Pressures from Consumption in the European Food System

**Philipp Schepelmann** [1,*] **, An Vercalsteren** [2] **, José Acosta-Fernandez** [1] **, Mathieu Saurat** [1] **, Katrien Boonen** [2] **, Maarten Christis** [2] **, Giovanni Marin** [3] **, Roberto Zoboli** [4] **and Cathy Maguire** [5]

[1] Wuppertal Institut für Klima, Umwelt, Energie gGmbH, Döppersberg 19, 42103 Wuppertal, Germany; joseacosta@wupperinst.org (J.A.-F.); mathieu.saurat@wupperinst.org (M.S.)

[2] Flemish Institute for Technological Research, Boeretang 200, 2400 Mol, Belgium; an.vercalsteren@vito.be (A.V.); katrien.boonen@vito.be (K.B.); Maarten.Christis@vito.be (M.C.)

[3] Department of Economics, Society, Politics, Università di Urbino Carlo Bo, 61029 Urbino, Italy; giovanni.marin@uniurb.it

[4] Department of International Economics, Institutions and Development, Università Cattolica del S. Cuore, 20123 Milan, Italy; roberto.zoboli@unicatt.it

[5] European Environment Agency, Kongens Nytorv 6, 1050 Copenhagen, Denmark; cathy.maguire@eea.europa.eu

* Correspondence: philipp.schepelmann@wupperinst.org

**Abstract:** The paper provides an integrated assessment of environmental and socio-economic effects arising from final consumption of food products by European households. Direct and indirect effects accumulated along the global supply chain are assessed by applying environmentally extended input–output analysis (EE-IOA). EXIOBASE 3.4 database is used as a source of detailed information on environmental pressures and world input–output transactions of intermediate and final goods and services. An original methodology to produce detailed allocation matrices to link IO data with household expenditure data is presented and applied. The results show a relative decoupling between environmental pressures and consumption over time and shows that European food consumption generates relatively less environmental pressures outside Europe (due to imports) than average European consumption. A methodological framework is defined to analyze the main driving forces by means of a structural decomposition analysis (SDA). The results of the SDA highlight that while technological developments and changes in the mix of consumed food products result in reductions in environmental pressures, this is offset by growth in consumption. The results highlight the importance of directing specific research and policy efforts towards food consumption to support the transition to a more sustainable food system in line with the objectives of the EU Farm to Fork Strategy.

**Keywords:** food consumption; environmentally extended input–output analysis; international trade; consumption-production perspective; structural decomposition analysis; value chain analysis; ex-post times series analysis; allocation tables

---

## 1. Introduction

Food is a basic human need but overconsumption, scarcity and insecurity can jeopardize health and quality of life. The food system is a complex global network of production, consumption and trade and is shaped by many factors: economic, environmental, political, technological and social, including cultural norms and lifestyles [1]. The food system inextricably links human health and social wellbeing with environmental sustainability. The EAT-Lancet Commission on Food, Planet,

Health highlighted how sustainable diets have lower environmental impacts and contribute to food and nutrition security [2].

The transition to a more sustainable food system is receiving increasing policy focus as reflected in the European Union's recent Farm to Fork Strategy under the European Green Deal. Currently, the food system is responsible for a range of impacts on the environment through emissions of pollutants, depletion of resources, waste generation, loss of biodiversity and degradation of ecosystems in Europe [1,3–5]. Europe is also highly dependent on imported final and intermediate products to satisfy European domestic demand for food with trade resulting in negative impacts outside Europe [3]. Many studies have identified the important role of consumption of food and beverages in terms of generating environmental pressures. More specifically, results based on single-country environmentally extended input–output analysis (EE-IOA) for nine EU member states suggested that in 2005 final consumption in the 'food area' contributed to 21% of greenhouse gas emissions, 49% of acidifying emissions, 20% of ground ozone precursors and 40% of total material requirement [6]. Analysis using a multi-regional input–output (MRIO) model based on data from EXIOBASE 2.2 estimated that food consumption in 2007 in the EU28 was responsible for 9.5% of the carbon footprint, 51.1% of the land footprint, 26% of the material footprint and 60.7% of the water footprint [7]. Life cycle assessment (LCA) approaches used to calculate the environmental footprint of food consumption for the EU28 for 2010, with environmental impact categories with impacts broken down by food products, highlighted the large contribution of meat and dairy products to environmental pressures [8].

This paper presents an integrated assessment methodology based on EE-IOA to assess direct and indirect environmental and socio-economic effects arising from food consumption. By taking a 'consumption perspective' in an EE-IOA framework, the paper provides measures of international transfer of environmental and economic effects along the global food value chain. The assessment uses EXIOBASE 3.4 as a source of data and covers all the consumption categories within the food system. The assessment is based on the development of detailed allocation matrices, which enable the calculation in a rigorous way the environmental pressures and impacts of consumption patterns by linking the data on final consumption expenditure to EE-IO data. The procedure to develop the allocation matrices is presented in a fully transparent and detailed manner thus contributing to methodological improvements in integrated assessment. The paper also develops a decomposition analysis to measure the role of technology, consumption mix, and consumption level in driving the environmental and economic effects of food consumption. The analyses presented in the paper can support policy and decision making by providing detailed analytical knowledge on those sectors and environmental dimensions that should be targeted to reduce the environmental pressures of the food system.

## 2. State of the Art

A range of bottom-up and top-down approaches have been used to analyse the environmental impacts of human food consumption, including quantification of overall pressures and the impacts resulting from specific food consumption habits in households. The results of methods such as Life Cycle Assessment (LCA), input–output analysis (IOA) and other hybrid methods combining both LCA and IOA are different and not attributable solely to the characteristics of the methods. Other aspects that define the scope of the analysis, such as the geographical region, the period, the system boundaries, and the attention paid to comparing different bundles of consumed food also explain the different results.

There are a range of LCA-based studies on the environmental impacts of food consumption and nutrition [9]. The EU Joint Research Centre has taken an LCA approach to the development of Basket-of-Products (BoP) indicators, including food [6]. Although the estimation of the effects of household's consumption expenditure through IOA is based on fairly aggregated information, the availability of more detailed information on the products consumed by households enables analysis of more specific groups of products [7,10–12]. Studies have used both methodological approaches,

for example, to calculate the impact of food consumption by Swiss households [13]. The results of IOA and LCA of different European household activities including food consumption have been compared [14].

As a bottom-up approach, LCA makes it possible to quantify and evaluate in detail the different pressures and environmental impacts associated with the various processes needed to produce a given functional unit (a single product or production process). However, LCAs have strict system boundaries often neglecting a significant share of indirect impacts on the economy and the environment. As a top-down approach, MRIO analysis allows from a technical-economic point of view the estimation of the total environmental pressures and impacts induced by the consumption of a given product group along the global supply chain of all required inputs for their production. This estimation is based on information on the different monetary transactions observed in the global market between the economic activities of each country or region of the world as well as on the direct environmental pressures resulting from production activities in each of the sectors involved in the whole economic process. By using the Leontief model, moreover, there is no need to set system boundaries as all direct and indirect technical-economic requirements (and hence associated environmental effects) are considered. MRIO can also account for technological shifts in the production of intermediate inputs sourced from different countries.

In addition to these methodological differences, there are other aspects that determine the quality of the results and their subsequent interpretation. These aspects are related to the level of detail of the information available to perform both the calculations and the analysis. In the case of MRIO, one of the most important aspects is related to the allocation of the calculated effects to the different purposes, in terms of final use. The type of assessment presented in this paper requires the use of a classification of individual consumption according to purpose (COICOP) rather than a classification of products produced for consumption (CPA). This enables distribution of the environmental effects caused by the production of consumer goods among the different categories of specific consumption, as well as the analysis of the effects along the supply chain associated with changes in the consumption mode of a given product (e.g., more wood as heating fuel than wood in the form of furniture). Using a COICOP-based classification allows to account for alternative ways of satisfying the same consumer need, i.e., different mixes of (CPA or NACE) products and sectors. This allocation is typically defined in so-called allocation tables.

## 2.1. Allocation Tables

In general, IO tables represent the structure of production and consumption activities within the economy. Since production and consumption are determined by the use of products, the structure of the economy can be represented either according to the economic activities using and producing the products or according to the product groups produced and used in the economy. When IO analysis is used to study environmental pressures and impacts of specific consumption patterns, the data on final consumption expenditure by households presented in the IO tables (IOT) need to be linked to household expenditure data. Detailed data on goods and services purchased by households are organized according to the COICOP while the data on final consumption expenditure by households presented in the IO tables are organized according to product groups produced in the economy. The link between both is made via allocation tables or correspondence matrices, which are compiled on the basis of the product classifications underlying the IOT and COICOP.

Allocation tables allow a preliminary assignment of the several product groups represented in the IOT as used by households to the several product groups detailed in the COICOP classification. This aims to attribute the amount of private household expenditure for each of the several product groups to a certain product group consumed with a defined purpose. Depending on the group of products, the attribution of household expenditure to the several COICOP categories can be exclusive or multiple. In the case of the expenditure for product groups that are consumed for satisfying multiple purposes, additional steps are required for an accurate attribution. Due to the conceptual differences

that characterize the existing data sets, additional adjustments are necessary (price transformation). All these additional methodological steps are summarized in the "allocation tables", which detail the attribution of household consumption expenditure in terms of product groups to the COICOP categories of consumed products.

The related literature describes two procedures for the elaboration of allocation tables. In the first approach, each product group represented in the IO table is connected to a single category of consumed products (one-to-one allocation). This allows to keep the level of product groups used in the calculations. In these cases, in which a many-to-one assignment is required, several product groups are first aggregated to a single one before they are related to a particular COICOP category of products consumed. This procedure simplifies the construction of the allocation table but reduces the level of detail. In the second approach, a product group represented in the IO table is associated, if necessary, with different categories of consumed products by households. This more detailed allocation is referred to as a one-to-many allocation. By applying this procedure, a single product group can be allocated to a single category or distributed among different categories of consumed products. For example, the CPA product group 'textiles' can be allocated to the COICOP categories 'clothing' as well as 'furniture.' The CPA group 'glass products' can be associated with the COICOP categories 'construction', 'furniture' and 'packaging'.

From a theoretical point of view and because most groups of products produced in the economy are consumed for different purposes, applying the allocation approach one-to-many would allow to generate allocation tables more accurately. However, this implies the use of bottom-up data from Household Budget Enquiries (HBE), data on trade and transport margins, on value-added taxes, taxes and subsidies on products, etc. In most cases, limited data availability prevents the development of allocation tables at such detail. However, for some countries a one-to-many bottom-up approach has been used to compile a detailed allocation matrix. In the Netherlands, for example, [15] an allocation table was developed to decompose the total price paid by consumers into producer price, trade and transport margins, and value-added taxes. The update of this table [16] used detailed information from Statistics Netherlands. This enables, on the one hand, linking of the data on consumed products represented in the Budget Survey to the product groups represented in the supply table of the System of National Accounts (SNA), in which they are clustered into functional domains that differ from the COICOP categories. On the other hand, the used data enable the decomposition of the consumer price into basic price, taxes and subsidies on products, and trade and transport margins. In Germany, a similar method was applied using as classification of the consumption categories the SEA (Systematisches Verzeichnis der Einnahmen und Ausgaben der privaten Haushalte), which is the German implementation of the international COICOP standard [17]. The Federal Statistical Office has published a consumption allocation table which allows a conversion from CPA to COICOP. The Federal Statistical Office provides upon request a table with the trade margins and taxes on products for each product group in order to convert consumer prices (used to measure consumption expenditures) into producer prices (used to measure monetary flows represented in the input–output tables). Another example of such bottom-up allocation is the procedure applied for Flanders [18]. The Flemish IO-table is based on specific monetary and environmental data for Flanders and is part of an interregional IO-table, in which trade with the Brussels region and Wallonia is represented. A link to Exiobase datasets enables the quantification of import flows from outside Belgium. The household's final consumption vector in the IO table is disaggregated into different COICOP categories using a matrix in which the consumption by households is represented in purchase prices. In this way the COICOP categories are linked to the output of the sectors represented in the IO-table according to the NACE classification. The allocation table is developed bottom-up by the Federal Planning Bureau and is structured according to the different COICOP categories attributed to the NACE sectors. Bottom-up data from the Household Budget Enquiry (HBE) for Belgium [18], tax data, VAT data, trade and transport margins are used to generate the allocation tables. For example, trade margins are allocated to the trade sectors and the product value is allocated to the respective producing economic sectors.

Many examples of one-to-one or many-to-one allocation tables are available in the literature, as this approach requires a limited amount of data and is thus frequently used. For example, a many-to-one allocation table for CPA product groups to COICOP categories was developed applying a pragmatic procedure based on available data [6]. Most CPA two-digit product groups could be attributed directly to a defined COICOP category. However, missing data, e.g., from household surveys, as well as data on trade margins for a transformation of the pricing from basic to purchaser prices restricted a more detailed allocation. Thus, a few modifications in the coverage of certain COICOP categories had to be made or proxy COICOP categories were used to try to establish an imperfect but reasonable match with the given CPA product groups. For instance, in order to assign the CPA product group "food and beverages" to a COICOP category, the coverage of the original COICOP group 'Food and non-alcoholic beverages' (COICOP 01) was expanded to 'Food and beverages', i.e., including alcoholic beverages. The CPA product group 'leather and leather products' was fully assigned to COICOP category 'clothing and footwear', even though there are also other significant uses of leather in other COICOP categories. Since the 2-digit, upper level COICOP categories were the only ones that could be used, the allocation resulted in an aggregation of COICOP categories (1 digit).

The detailed allocation tables for a specific country are sometimes extrapolated to other European countries, for example, the allocation tables for the EU27 compiled based on an allocation table (in purchaser prices) for Austria [19]. For that, a correspondence matrix between the classification of supply and use tables (CPA 2002, 2-digit) and the COICOP categories was used. The authors used the RAS method for constructing the allocation matrices for the other countries based on COICOP data (in purchaser prices) from Eurostat. Additional statistical sources allowed to split the energy sector into electricity and heating. Electricity expenditure was estimated by combining IEA energy balance data (for the household sector) with IEA energy prices. Energy efficiency indices for heating and electrical appliances were taken from the ODYSSEE database, while the TREMOVE database was the primary source of the energy efficiency index for vehicles. The RAS method was also applied to construct allocation matrices for other countries, based on the 2004 German allocation matrix [20]. The author discusses the limitations of the approach, which implies that households of all countries use the same technology of converting industry products to goods. Not only may the countries be heterogeneous, but the weights may change in time. However, missing data for most countries may justify the use of the RAS method.

For some countries, allocation tables are elaborated using country-specific data on household consumption and on price components from national statistical offices. For Europe, both a simplified procedure linking one CPA to one COICOP and a more complex procedure applying RAS to extrapolate data from a specific country to other countries have been applied [6,19,20].

## 2.2. Limitations of Allocation Tables

Allocations of household consumption expenditure to COICOP categories are likely to change over time. The use of static allocations to calculate future scenarios can lead to potential inaccuracies [21]. Many recent studies use static coefficients [17,22–24]. However, technological change may lead to growth, reduction or substitution effects. Furthermore, the relative prices of goods may change significantly over the years. Assumptions on these dynamics are, in particular, relevant for forecasting and future scenario construction. For example, at the turn of the century, the share of air transport significantly increased at the expense of water transport. Air transport has an environmental impact substantially larger than water transport. Failing to account for this change in the transport mode would result in an underestimation of environmental impacts. There is no generally accepted approach or convention to construct allocation tables. The result is a multitude of different approaches. However, very few studies explain the underlying assumptions which were made for the construction of their allocation matrices.

As far as analyses for the food system are concerned, the literature indicates that the allocation of trade activities is a major issue. When the IO tables are in basic prices, trade margins require

a correct allocation to the different COICOP categories for generating reasonable allocation tables. These trade margins include the (impact of) trade and retail (e.g., cooling, transport, warehousing), which contribute substantially to environmental pressures related to the food system. Ideally the food system is not restricted to COICOP 01 'food' but should also include COICOP 11.1 'catering' and other relevant COICOP categories. A clear definition of the food system is thus very important.

## 3. Research Questions

The objective of the analysis is to provide a detailed assessment of the different environmental pressures and some relevant socioeconomic effects caused by food consumption in the European Environment Agency (EEA) member countries (member countries at the time of analysis were the EU28 plus Norway, Iceland, Switzerland, Lichtenstein and Turkey. The geographical scope was selected to ensure that the analysis covered the largest share of EEA countries' economies as possible. In terms of illustrating the method and the key results and conclusions, the inclusion of non-EU countries has not significantly affected the results as they align well with other EU28 estimates). The assessment focused on the following research questions:

(i)   Are European households changing their food consumption habits to goods and services that generate less environmental pressure?

(ii)   What are the environmental and socio-economic effects in Europe and in the rest of the world resulting from food consumption in European households?

To carry out this analysis, COICOP was used rather than CPA to enable the distribution of the environmental effects caused by the production of consumer goods among the different categories of specific consumption, as well as analysis of the effects along the supply chain associated with changes in the consumption of a given product (e.g., more wood as heating fuel than wood in the form of furniture).

Furthermore, applying detailed environmentally and socioeconomically extended multi-regional input–output models (ESE-MRIOM), which are built from corresponding extended multi-regional input–output (MRIOT) tables, considerably increases the scope of the analysis. An ESE-MRIOM is a powerful analytical instrument for assessing different pressures and effects of production and consumption activities from a system perspective. It enables the identification of the different exporting economies in which environmental pressures and socioeconomic effects are induced by domestic consumption. Thus, the estimation of total pressures and effects occurs with greater geographical precision not only in terms of the origin, quantity and mix of products consumed, but also in terms of the technology used for their production. In this way, the identification of the hotspots of the different environmental and socioeconomic effects is characterized by a high degree of detail.

The European Topic Centre on Waste and Materials in a Green Economy has already used the ESE-MRIOM for various analytical purposes including estimating the environmental impacts of final demand in EEA member countries [1]. However, the analytical spectrum of ESE-MRIOM goes beyond this. An example of these other applications for the analysis of the food system is the structural decomposition analysis (SDA). The application of the SDA allows identification and quantification of the contribution of the main driving forces of changes in observed environmental pressures, among others, associated with food consumption. This information is important, because it indicates possible intervention points for policies aimed at reducing the environmental pressures related to food consumption.

## 4. Methods and Data

The method and data section first describes the EXIOBASE database used throughout this paper. Secondly, the definition and scope of the food system as applied in this paper is discussed, followed by the description of the construction of the allocation matrix. Finally, the methodologies for the input–output calculations and for the SDA are described.

## 4.1. Exiobase v3.4

EXIOBASE (v3.4) is a database containing detailed information on the world economy, the monetary flows associated with the supply and use of goods and services produced and consumed and the direct environmental and socio-economic effects of production and consumption activities [25]. The monetary flows are organized in the form of Multi-Regional Supply and Use/Input–Output Tables (MRSUT/MRIOT) with a defined industry and product group structure. The environmental and socio-economic effects are structured in the form of multiregional matrices with the same industry structure as the MRSUT/MRIOT. EXIOBASE was developed by detailing and harmonizing country-specific monetary SUT (MSUT) and data related to energy, emissions, water use, land use, resource extractions as well as employment and value added by industry. The country-specific MSUT is linked via trade and extended with environmental and socioeconomic variables (ESE-MRSUT), on which an environmentally and socioeconomically extended multi-regional input–output table (ESE-MRIOT) is built. The ESE-MRIOT can be used for an analysis along the supply chain of the environmental pressures and socio-economic effects associated with the final consumption of product groups. This version of EXIOBASE is a time series of ESE MRIOT ranging from 1995 to 2011 for 44 countries (28 EU member plus 16 major economies) and five regions of the rest of the world. The distinguishing characteristics of EXIOBASE are the high level of consistent sectoral (200 products, 163 industries for all countries and regions included) and environmental detail. For the analysis carried out here, the product-by-product tables of EXIOBASE were used.

Compared to single-country studies, the use of detailed ESE-MRIOT from EXIOBASE enables a more precise identification of the different components that contribute to (changes in) aggregate environmental or socio-economic effects. This enables the identification of policy-relevant hotspots at a detailed level.

## 4.2. Definition of the Food System

Definitions of the food system tend to be broad and can include all the elements (environment, people, inputs, processes, infrastructures, institutions, etc.) and activities that relate to the production, processing, distribution, preparation and consumption of food, and the outputs of these activities, including socio-economic and environmental outcomes [3]. However, the applied data and models require a definition of the food system in terms of COICOP categories. Table 1 presents the two-digit COICOP categories. The food system as defined in this paper includes the COICOP categories marked Italic, i.e.:

01.1 Food
01.2 Non-alcoholic beverages
02.1 Alcoholic beverages

**Table 1.** Definition of food system in terms of classification of individual consumption according to purpose (COICOP) categories [25].

| COICOP | | | Description |
|:---:|:---:|:---:|:---:|
| 2-digit | 3-digit | 4-digit | |
| 01 | | | Food and non-alcoholic beverages |
| | 01.1 | | Food |
| | | 01.1.1 | Bread and cereals |
| | | 01.1.2 | Meat |
| | | 01.1.3 | Fish |
| | | 01.1.4 | Milk, cheese and eggs |
| | | 01.1.5 | Oils and fats |
| | | 01.1.6 | Fruit |
| | | 01.1.7 | Vegetables |
| | | 01.1.8 | Sugar, jam, honey, chocolate and confectionery |
| | | 01.1.9 | Food products n.e.c. |
| | 01.2 | | Non-alcoholic beverages |
| 02 | | | Alcoholic beverages, tobacco and narcotics |
| | 02.1 | | Alcoholic beverages |
| 03 | | | Clothing and footwear |
| 04 | | | Housing, water, electricity, gas and other fuels |
| 05 | | | Furnishings, household equipment and routine household maintenance |
| 06 | | | Health |
| 07 | | | Transport |
| 08 | | | Communications |
| 09 | | | Recreation and culture |
| 10 | | | Education |
| 11 | | | Restaurants and hotels |
| 12 | | | Miscellaneous goods and services |

In our allocation tables, the COICOP categories 01.2 and 02.1 are aggregated because the resolution of the product group nomenclature in EXIOBASE does not allow this distinction. The COICOP category "catering" (11.1) should ideally be included in the food system, however, in the EXIOBASE nomenclature the product group "Hotel and restaurant services", which includes catering, cannot be further disaggregated. For this reason, the analysis focusses only on the environmental pressures and socio-economic effects caused by the use and preparation of food products by private households. We acknowledge that further product groups or services should at least partly be included (e.g., energy, water, hotel and restaurant services) but cannot be allocated due to lack of detailed data. Possible over- or underestimations will be addressed in the discussion.

*4.3. Construction of the Allocation Matrix*

For analysis of consumption systems based on the purpose for which products are used rather than on total quantity of product groups produced for consumption in general, the establishment of the correspondence between the product classification inherent to the input–output tables used (EXIOBASE is based on CPA 2002) and the COICOP classification is required. Eurostat publishes two correspondence tables: CPA 2002 to COICOP 1999 and CPA 2008 to COICOP 1999. These correspondences help to generate the allocation tables that allow attributing the results of input–output modeling, which are structured according to the groups of products produced, to the COICOP consumption categories.

Since the research questions focused on the effects associated with the different purposes for which households consume food products, it was necessary to construct allocation tables. The procedure applied for this is summarized in the following steps:

First, we identified products exclusively for technical/industrial use or not concerning the consumption activities of households (e.g., p21.1 paper pulp) among the list of 200 goods and services of the EXIOBASE classification. This identification makes use of the Eurostat correspondence tables at the six-digit level of the CPA classification. In some cases, we assigned a product group to household

consumption even though the product group should be excluded (e.g., p24.g bio gasoline). The reason is that for such product groups the household consumption column in the input–output table of EXIOBASE does contain entries. Second, we clustered the product groups that cannot be assigned to COICOP categories. Third, we had to make decisions on how to allocate the household consumption product groups to the corresponding COICOP categories. There are three cases:

1. If there is an unambiguous one-to-one correspondence between a product group of the EXIOBASE classification and a COICOP category, we assigned this product group to this one COICOP category only. For example, the product p15.b."Products of meat" is unambiguously attributed one-to-one to the COICOP category 01.1.2 "Meat".

2. If there is an unambiguous one-to-many correspondence between a product group of the EXIOBASE classification and several COICOP categories at the 4-digit level that all belong to the same category at the 3-digit level, we created a combined COICOP category at the 4-digit level and assigned the product group to that new category. For instance, the EXIOBASE products P01.d. "Vegetables, fruits, nuts" corresponds to COICOP categories 01.1.6 "Fruit" and 01.1.7 "Vegetables" and we created a combined COICOP category at the four-digit level 01.1.6_01.1.7 "Fruit and Vegetables".

3. If there is an ambiguous one-to-many correspondence between a product group of the EXIOBASE classification and several COICOP categories that cannot be reconciled as in case 2, we arbitrarily assigned the EXIOBASE product group to one COICOP category, thus turning the one-to-many into an arbitrary one-to-one attribution.

Note that case 2 and 3 may actually overlap. For instance, EXIOBASE p17 "Textiles" corresponds to the COICOP categories 03.1 "Clothing", 05.1 "furniture and furnishings, carpets and other floor coverings", 05.2 "household textiles" and 09.3 "other recreational items and equipment, gardens and pets." We arbitrarily assigned the EXIOBASE product group p17 to a new COICOP category combining 05.1 and 05.2.

Assigning an entire product group to a single COICOP category leads to over- and underestimates when the input–output analysis is carried out. On the one hand, the environmental pressure generated by the COICOP category to which all the consumption of the relevant EXIOBASE product group has been attributed will be overestimated. On the other hand, the environmental pressure of the other relevant COICOP categories from which the product group has been subtracted will be underestimated. However, country-specific additional information on the CPA 2002 classification that would allow estimating the shares of household expenditure for the several products aggregated in each EXIOBASE product group is not available. In the case of the EU Member States, such data provided by Eurostat are available only at the 2-digit level of the CPA 2008 classification and are therefore useless for a breakdown.

At the end of this process we generated an allocation table which includes:

- The unambiguous allocations of all one-to-one and one-to-many correspondences (case 1 and 2).
- The one-to-many attributions are all arbitrarily turned into one-to-one attributions, as described under case 3. For instance, the EXIOBASE product group "Products of forestry, logging and related services" (18 p02) belong to COICOP categories "Solid fuels" (04.5.4) as well as "Gardens, plants and flowers" (09.3.3). Allocation Table 1 attributes the "Products of forestry, logging and related services" fully to COICOP "Solid fuels" (04.5.4).
- The product groups that cannot be assigned to COICOP categories are clustered in a new COICOP category "Miscellaneous goods and services (not applicable for private households)" (12.) Such an EXIOBASE group is, for example, "iron ores" (33 p13.1).
- Some product groups from untypical clusters were assigned to new n.e.c. (i.e., not elsewhere classified) categories within the existing COICOP domains. This approach allows to keep them separate and to treat them analytically like category 12 describe above. For instance, EXIOBASE

"Sugar cane, sugar beet" (6 p01.f) is assigned to a new COICOP category "01 Food > 1.2 Food n.e.c. > 01.2.0 n.e.c."

- The EXIOBASE product groups "Wholesale trade and commission trade services, except of motor vehicles and motorcycles" (154 p51) and "Retail trade services, except of motor vehicles and motorcycles; repair services of personal and household goods" (155 p52) are allocated to all COICOP categories according to the share of each COICOP category in the final consumption expenditure of households. This approximation assumes that the share of the trade margins in the final consumption expenditure of households is the same (or very similar) across all COICOP categories. We use the Eurostat dataset on Final consumption expenditure of households by consumption purpose (COICOP 3 digit) [nama_10_co3_p3] for this approximation.

- The EXIOBASE product groups "Railway transportation services" (157 p60.1), "Other land transportation services" (158 p60.2), "Sea and coastal water transportation services" (160 p61.1), "Inland water transportation services" (161 p61.2), as well as "Air transport services" (162 p62) are allocated to all COICOP categories dealing with goods (i.e., services are here excluded because the term "transportation" in general is rather applicable for transport of goods and persons but not for services, note that we included repair activities of goods) according to the share of each COICOP category in the final consumption expenditure of households [26]. We used the same Eurostat dataset as above for this approximation. The allocation of the remaining items (i.e., the total shares of services in the final consumption expenditure of households) is entirely assigned to the corresponding subcategories of the COICOP category "Transport" (07). For instance, EXIOBASE "Railway transportation services" (157 p60.1) is allocated to all COICOP categories dealing with products (not services) using the Eurostat data and the rest is assigned to the COICOP category "Passenger transport by railway" (07.3.1). This relies on the assumptions that (1) the shares of the transport margins in the final consumption expenditure of households is the same (or very similar) across all COICOP categories dealing with products (services do not require transport); and (2) the private household final consumption of transport services in IOT in basic prices (such as EXIOBASE) is made of both accumulated transport margins and the direct consumption of transportation services.

*4.4. Extended Input–Output Analysis*

Extended input–output analysis is applied to provide the results on the ex-post time series analysis and the value chain analysis. The global pressures and effects (footprints) associated with final consumption of food products have been calculated with an extended multiregional Input-Model built from EXIOBASE data [27]. To this end, environmentally and socio-economically extended product-by-product tables were used. The core equations of the model are as follows:

$$x = A \cdot x + y \tag{1}$$

where $x$ is the total output vector, $A$ is the matrix of direct input coefficients (also referred to as the matrix of technological coefficients), and $y$ is the final demand vector. Solving the equation for output transforms it into [28]:

$$x = (I - A)^{-1} \cdot y = L \cdot y \tag{2}$$

where $I$ is the identity matrix, and $L$ is the Leontief inverse also referred to as matrix of direct and indirect output requirements per unit produced for final demand or, more simply, multiplier matrix. The Leontief model implies the following assumptions [29]: prices are fixed in the short term, input coefficients are constant regardless of output or final demand level changes, structure of the economy is taken to be constant, at least in the reported period.

The direct environmental and socio-economic effects of national production are by definition the result of the sum of the direct effects associated with each unit produced in each industry:

$$E^T = \sum_1^n E_i = \sum_1^n e_i^{int} \cdot x_n = \left\langle e^{int} \right\rangle \cdot x \tag{3}$$

where $E^T$ is the total environmental or socio-economic effect associated with the corresponding amounts of the final output $x$ and $e^{int}$ is the environmental or socio-economic effect intensity vector. Each element of $e^{int}$ represents the amount of the effect directly caused by the production of a product group. $E^T$ is also what we call the effect measured from the production perspective.

By substituting the vector $x$ Equation (2) into Equation (3), an extended input–output model is created:

$$E^T = \left\langle e^{int} \right\rangle \cdot x = \left\langle e^{int} \right\rangle \cdot (I-A)^{-1} \cdot y \tag{4}$$

Applying Equation (4), the total footprint attributed to each of the different sectors of final demand is calculated. To carry out the calculation of the footprint related to each product group used to satisfy the final demand another expression of this model must be applied.

$$E^T = e^{int} \cdot x = e^{int} \cdot (I-A)^{-1} \cdot < y \geq = e^{acc} \cdot \langle y \rangle \tag{5}$$

where $e^{acc}$ is the environmental or socio-economic effect intensity accumulated along the whole supply chain. Because all direct and indirect input requirements per unit of product group produced are represented in the Leontief inverse (L), their multiplication with the vector of the direct intensity ($e^{int}$) leads to the calculation of the environmental and socioeconomic accumulated intensity of each product group. The environmental or socioeconomic accumulated effect intensity ($e^{acc}$) is also called environmental or socioeconomic multiplier or accumulated technological effect.

The equations above are still valid in a multi-regional model such as EXIOBASE. In cases such as these, $E^T$ simply consists of all individual country effects. We will aggregate individual country environmental or socio-economic effects into EEA countries' footprints.

The input–output calculations will generate measurements of footprints for the selected environmental and socio-economic effects associated with the final consumption of European private households. Such footprints measurements include not only production effects but also effects resulting from the ultimate use of the various products (e.g., the amounts of $CO_2$ resulting from gasoil production as well as from gasoil combustion). There will be a footprint for each product category available in the EXIOBASE nomenclature, distinguishing products domestically produced and imports. Each domestic and imported footprint will then be distributed to consumption purpose categories using the allocation tables described in Section 4.3. The operation is an element-wise multiplication of the environmental footprint vector ($E^T$) defined in Equation (5) with the "product-x-coicop category" allocation table ($T^{alloc}$) resulting in a ($E^{alloc}$) matrix as follows:

$$E_{ij}^{alloc} = E_i^T \cdot T_{ij}^{alloc} \tag{6}$$

*4.5. Extrapolation*

The extended input–output analysis using Exiobase v.3.4 results in ex-post time series results for pressures and effects for the period 1995–2011. A calculation of pressures and effects for the year 2017 is added via an extra extrapolation step. This extrapolation makes use of household expenditure data available from Eurostat (final consumption expenditure of households by consumption purpose (COICOP 3 digit) [nama_10_co3_p3]), which are corrected for inflation (HICP (2015 = 100)—annual data, average index and rate of change [prc_hicp_aind]), and the trend in intensities of the different pressures and effects from Exiobase. We used a linear trend based on the 1995–2011 data to estimate the 2017-intensities for household consumption of food products, total household consumption and total

final demand. The linear trend is calculated through a given set of dependent y-values (i.e., intensities) and a set of independent x-values (i.e., years) and return values along the trend line, making use of the least squares method. Multiplying these extrapolated intensities with expenditures in constant prices results in total pressures and effects. Because the 2017-intensities are a result of the extrapolation, the analysis on intensities is limited to the 1995–2011 period. The 2017-footprint should be interpreted with caution, due to the uncertainty in forecasting the intensities.

### 4.6. Structural Decomposition Analysis

A change of a variable is a result of changes in the determinants of the variable. For instance, the production technology and the volume of final demand determine a change of the macro-economic indicator "gross production." Currently, two main methods are applied in the relevant literature to evaluate the contribution of determinants: SDA and Index Decomposition Analysis (IDA). Both methods allow to quantify the positive and negative contributions of each determinant over time. The historical variation of these determinants in turn results in the dynamics of change of macro indicators such as employment, value added, $CO_2$ emissions, etc. SDA uses input–output tables, IDA uses aggregate data at the sector-level. This analysis presents the results of an SDA.

The magnitude of the environmental and socio-economic effects resulting from production and consumption activities in an economy depends on:

(a) the direct effect per unit of a produced output,
(b) the technology applied in each sector, and
(c) the total volume and composition of final consumption.

Consequently, a structural decomposition of environmental and socio-economic effects of domestic demand basically is based on the quantification of three determinants of change:

- intensity effect (changes in the direct effect per unit output)
- technological effect, (changes in total requirements of intermediate per unit output)
- final demand effect (changes in total output volume and composition of final demand)

Formally, the quantification of the contributions is derived from the algebraic formula by which the environmentally extended input–output model is represented, as explained in Equation (4). However, those contributions can be reformulated when the following two aspects are taken into account. First, the final demand is determined by the absolute consumption volume as well as the structure or mix of consumed product groups. Therefore, it can be further disaggregated into both determinants. Second, both the contribution of the observed changes in the composition of the intermediate inputs required for production by each sector (Leontief-multiplier), as well as the contribution of the changes in the direct effect per unit produced in each sector (direct intensity effect) represent technological effects. Since these contributions are determined by the production characteristics prevailing in the industries of the economies from which the products originate, both "technology" contributions can be combined into a single determinant.

Rearranging the determinants this way, the quantification of the corresponding contributions that explain the change of the analyzed parameter $\Delta E^T$ (e.g., environmental pressure, valued added, employment) between two points in time (0 and t) can be expressed in its additive form, as follows:

$$\Delta E_{0-t}^T = I_{contribution} + Y_{contribution}^{vol} + Y_{contribution}^{str} \qquad (7)$$

where $I_{contribution}$, $Y_{volcontribution}$ and $Y_{strcontribution}$ capture the "specific effects" in terms of accumulated technology effect ($e^{acc}$ in Equation (5) or direct and indirect effect intensity), final demand volume effect, and final demand structure effect, respectively. For instance, if the decomposition of observed change in environmental footprint $E^T$ caused by final demand in a defined geographical region and time period shows two specific effects with a positive value and one with a negative value, then effects

with a value greater than zero have contributed to increasing the environmental or socio-economic impact, while the effect with a negative value has contributed to reducing it. In this way, the net effect is the sum of the different effects.

There are a number of algorithms available to reach the decomposition laid out in Equation (6), which can lead to significantly different results, see e.g., [30–33]. For example, when assessing the contribution resulting from changes in applied technology, we could use either beginning-of-period or end-of-period environmental multipliers for the calculation, which would lead to different results. To overcome the non-uniqueness issue, we will use the algorithm that has the particularity to be the best approximation of the average of all possible decomposition forms [34]. The mathematical formulae and their full derivations are presented in an extensive review of decomposition methods [35].

## 5. Results

The results section firstly describes the results of the ex-post time series analysis. It describes the contribution of food consumption to the footprints of total household consumption and total consumption in terms of two socio-economic effects, four resource use categories and three environmental impact categories. The results of other consumption domains are shown in Appendix A.

Secondly, the value chain analysis shows the share of these footprints that is located outside EEA member countries. Finally, the SDA assesses the contributions of determinants of changes in environmental or socio-economical footprints induced by private household food consumption.

*5.1. Ex-Post Time Series Analysis*

Ex-post time series analysis is applied to assess the contribution of food consumption to total consumption of households in EEA member countries, in terms of environmental impact, value added and jobs, and how this evolved over time. From a policy perspective, it is relevant to know whether European households are switching consumption patterns to goods and services with fewer environmental pressures, and more specifically, towards a more environmentally favorable diet.

In order to carry out the mentioned analysis, footprints for different years were calculated. For that, the ESE-MRIOM or Equation (5) was applied and the available data time series were used. Figure 1 presents a time series of the different footprints of food consumption by households, total household consumption (not only food) and total final consumption (incl. consumption by government, investments by industry, etc.) Indicators are grouped according to their focus, i.e., socio-economic parameters, resources use and impacts:

- related to food consumption by private households (COICOP 01 + 02.1);
- related to private household consumption of all other product groups in other COICOP categories;
- related to non-household final demand (e.g., investments, government expenses).

The total 2017 value is shown, which is the footprint per impact or resources use indexed to 100 on the vertical axis.

While overall final demand (in constant prices) grew by about 37% from 1995–2017, the environmental footprints of final consumption and private household consumption grew less (Global Warming Potential (GWP), energy, material and water consumption) or even decreased (acidification, eutrophication, land use). This indicates relative decoupling between GDP and the GWP, energy, material and water footprints of final consumption; with absolute decoupling occurring between GDP and acidification, eutrophication and land use footprints. The expenditure share of household consumption in total consumption was 47% in 2017 and stable compared to the 1995 and 2005 shares. This share is higher for all impacts and resource use, meaning that household consumption generates on average more impact and requires more resources compared to average investment and government expenditure.

The value added specifically for food consumption of private households also follows this overall trend, increasing by 5% between 1995 and 2017. The environmental footprints of food consumption

by households increased much less, demonstrating relative decoupling with absolute decoupling for land use, acidification and eutrophication. In terms of expenditures (or gross value added) in 2017, the share of food consumption in total domestic final demand was 7% and its share in total household consumption was 14%. Compared to 1995 values, the shares slightly decreased, indicating that expenditure on food products by households has decreased in relation to total spending. For all impacts and resource use, the share caused by household food consumption in total household final demand was higher than the share of expenditures, except for energy use. The highest shares are for land use (71%), water consumption (61%), eutrophication (53%), acidification (48%) and material use (33%).

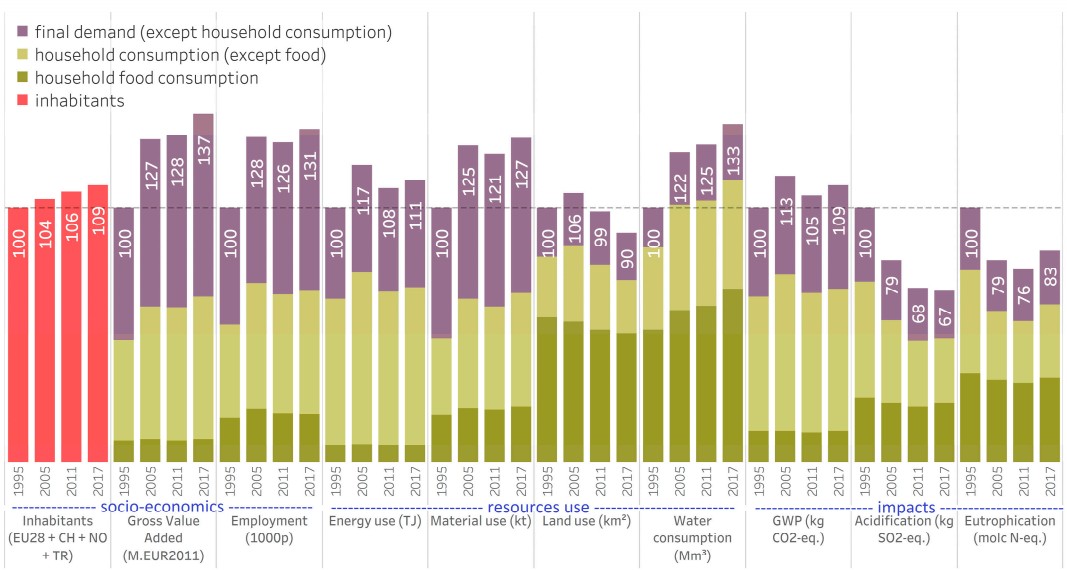

**Figure 1.** Impact and use of resources by consumption in EEA member countries, 1995–2005–2011–2017.

To assess if European households are switching consumption patterns to food products with fewer environmental pressures, thus towards a more environmentally favorable diet, we focus on the GWP, water and land use of household food consumption (Table 2). GWP remains more or less stable in total absolute terms although population increases, and thus shows a relative decoupling. However, absolute decoupling is close to being achieved as GWP showed a very small decrease between 1995 and 2017. The food products that contribute most to GWP are bread and cereals (01.1.1; 33–37%), meat (01.1.2; 25%) and milk, cheese and eggs (01.1.4; 17–21%), followed by fish (01.1.3; 8%) and fruit and vegetables (01.1.6 + 7; 8%). The product-specific global environmental pressures, i.e., $E_{ij}^{alloc}$ in Equation (6), were analyzed over a shorter time period (1995–2011) (Table 2). The discussion on the trend in intensities is limited to the period 1995-2011, as the 2017 intensity is an extrapolation result of this time period (see Section 4.5). For GWP, the most important changes occurred for milk, cheese and eggs (reduction of 23%), 'other' food products (increase of 16%), oils and fats (increase of 11%), fish (reduction of 11%), bread and cereals (increase of 5%) and meat (reduction of 3%) Looking at the total water use related to household food consumption, this volume increased significantly by 30% between 1995 and 2017. The food products that contribute most to the water footprint are bread and cereals (01.1.1; 50%), fruit and vegetables (01.1.6–01.1.7; 20–25%) and meat (01.1.2; 12%). The most important relative changes in water use intensities of food products for household consumption occurred for bread and cereals (increase of 25%), fruit and vegetables (increase of 18%), fish (increase of 16%), meat (increase of 14%) and milk, cheese and eggs (reduction of 11%). The land use related to household food consumption reduced by 11% between 1995 and 2017. The food products that contributed most to land use are bread and cereals (01.1.1; 40–46%), meat (01.1.2; 21%), fruit and vegetables (01.1.6–01.1.7; 13–16%) and milk, cheese and eggs (01.1.4; 9–12%). The most important changes in land use intensities of food products for household consumption occurred for milk, cheese and eggs (reduction of 31%),

fruit and vegetables (reduction of 20%) and fish (reduction of 19%). The land use related to meat was reduced slightly.

**Table 2.** Changes in household expenditure and intensity for different food products.

| Change between 1995 and 2011 | Household Expenditures | GHG Emission Intensity | Water use Intensity | Land Use Intensity |
|---|---|---|---|---|
| Bread and cereals (↑ 5% *) | ↓ 5% | ↑ 10% | ↑ 32% | ↑ 10% |
| Meat (↓ 3%) | stable | ↓ 5% | ↑ 12% | ↓ 8% |
| Fish (↓ 11%) | ↓ 5% | ↓ 5% | ↑ 25% | ↓ 13% |
| Milk, cheese and eggs (↓ 23%) | stable | ↓ 22% | ↓ 10% | ↓ 30% |
| Oils and fats (↑ 11%) | stable | ↑ 9% | stable | ↓ 29% |
| Fruit and vegetables (stable) | ↓ 10% | ↑ 11% | ↑ 31% | ↓ 12% |
| Sugar, jam, honey, chocolate and confectionary (↓ 6%) | ↓ 6% | stable | ↓ 23% | ↓ 43% |
| Food products n.e.c. (↑ 16%) | ↑ 57% | ↓ 26% | ↑ 12% | ↓ 19% |
| Beverages (↑ 1%) | stable | stable | ↑ 29% | ↑ 8% |

\* Percentages relate to change in total global warming potential 'impact' between 1995 and 2011.

The changes in impact are a combined effect of the share of the type of food products in the household's diet (expenditures) and the environmental pressures (GHG emission, water and land use intensity) caused along the production chain of the respective food products. The latter can be caused by different factors not analyzed here, e.g., a changed basket of products (e.g., switch from beef to chicken meat), improved production efficiency, etc.

*5.2. Value Chain Analysis*

A value chain analysis is used to identify the part of the footprint, calculated by applying the ESE-MRIOM or Equation (5), that is located outside EEA member countries providing insight into which part of the world and in which sectors food consumption of households is creating environmental pressures and impacts, value added and jobs, and how this has evolved over time. From a policy perspective, it is important to know whether Europe is shifting the environmental burden to other regions by the changing food consumption patterns, and to know the related benefits i.e., employment and value added.

Figure 2 shows that a substantial and increasing (except for eutrophication) share of global environmental and socioeconomic effects caused by total final demand in EEA member countries occurs outside these countries. The contribution of resources extracted or used outside EEA member countries to the footprint of food consumption is illustrated by the estimate that more than half of total requirements for land use (57–61%) and water consumption (52–59%) occurred overseas and these are strongly correlated with agricultural production. The 'benefits' related to the reliance on imports are much less: the value added created outside EEA member countries by final demand was only 7% in 1995, increasing to 11% in 2011 (the discussion on the proportion of EEA's final demand footprint is limited to the period 1995–2011, as this proportion is derived directly from the Exiobase-model). This follows a rising trend but remains low. The same applies to the jobs created abroad due to final demand, the overseas share increased from 37 to 46% in the same period. Together, the figures on value added and employment suggest that, overall, final consumption created employment in low-value-added activities overseas.

The share of global environmental pressures and impacts generated outside EEA member countries by food consumption of households also showed an increasing trend between 1995 and 2011 (Figure 3). However, the share of resource use and environmental impacts exerted overseas from food consumption is smaller than the share generated by total final demand, with the exception of energy use where the overseas share related to food consumption is higher than that related to total final demand. The share of value added and jobs generated outside EEA member countries by households food consumption is

larger than the share generated by total final demand. In 2011, 16% of the gross value added in the food production chain was generated outside EEA member countries (compared to 11% for total final demand) and 60% of the employment was located abroad. This means that food consumption in EEA member countries generates relatively less environmental impact abroad than average and creates relatively more value added there.

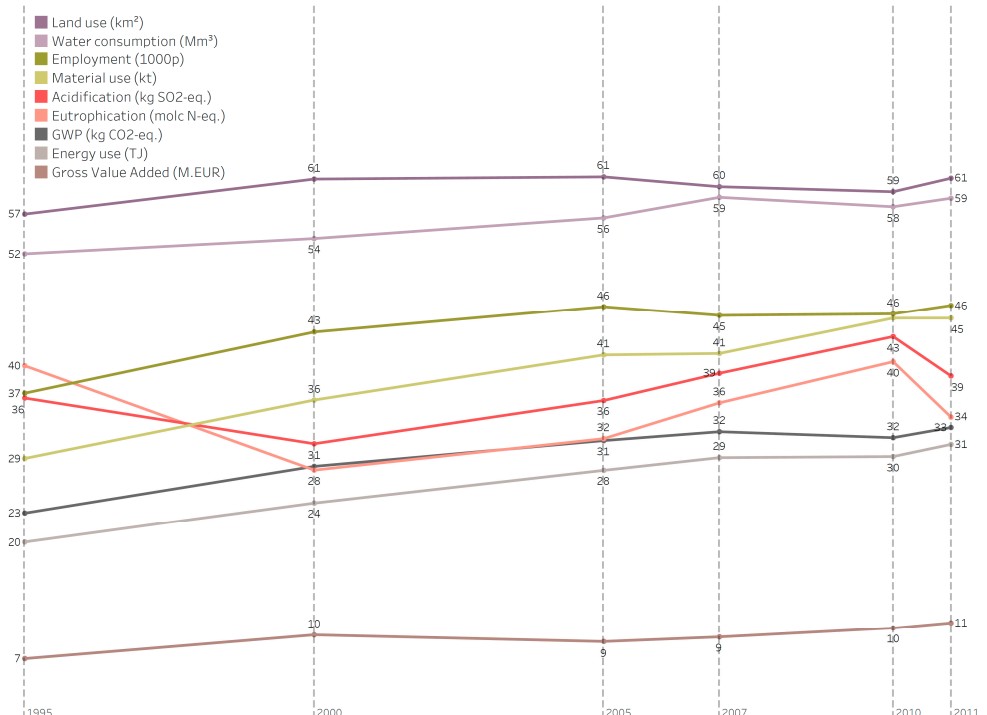

**Figure 2.** Proportion of EEA's final demand footprint exerted outside EEA's borders.

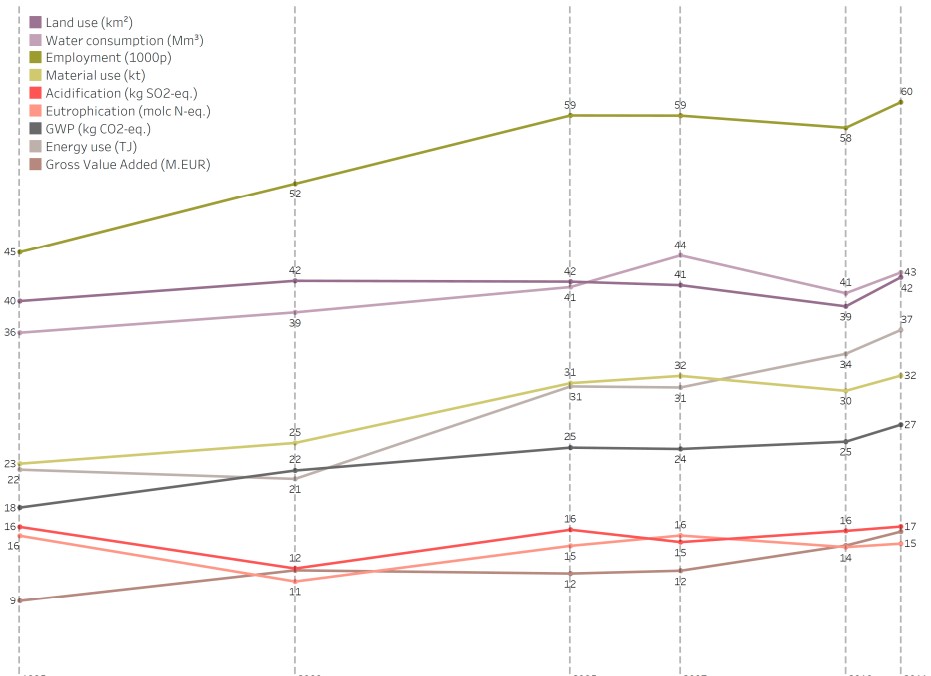

**Figure 3.** Proportion of EEA households' food consumption footprint exerted outside EEA's borders.

Table 3 below adds more regional detail to the share exerted outside Europe. Most of the value added by household food consumption is generated in Asia and the Pacific region (7%) along with most of the environmental impact and resource use, with the region's share increasing steadily from 1995 onwards.

**Table 3.** Share of impacts across world regions, generated by EEA households' food consumption. The geographical and sectoral distribution follow from the IOA, using Exiobase. Therefore, the 2011 results are presented and not the extrapolation value for 2017.

| Share in Geographical Regions (in 2011) | Food Consumption by Households in EEA Countries | | | | | | | | |
|---|---|---|---|---|---|---|---|---|---|
| | Gross Value Added | Employment | Global Warming Potential | Acidification | Eutrophication | Energy Use | Land Use | Material Use | Water Use |
| Europe | 84% | 40% | 73% | 83% | 85% | 63% | 58% | 68% | 57% |
| North America | 3% | 1% | 3% | 1% | 1% | 7% | 3% | 2% | 3% |
| South America | 3% | 5% | 4% | 4% | 4% | 2% | 10% | 8% | 9% |
| Africa | 2% | 26% | 4% | 3% | 2% | 3% | 13% | 7% | 11% |
| Asia and Pacific | 7% | 27% | 13% | 7% | 6% | 19% | 13% | 12% | 16% |
| Middle East | 2% | 2% | 4% | 2% | 1% | 7% | 2% | 3% | 3% |

The value chain analysis enables a more detailed examination of the industries/sectors where food consumption causes environmental impacts and creates value added (Table 4). In 2011, 52% of the gross value added in the food production chain was linked to agriculture and food manufacturing. The other 48% was distributed across services (16%), transport (11%), trade (10%), energy (4%), plastic and chemicals (2%), metals (2%), electronics (1%) and minerals (1%). The resource use and impacts are more concentrated in the agriculture and food manufacturing, except for energy use. The relative importance of the different sectors has remained more or less stable over time.

**Table 4.** Share of impacts across industries/sectors worldwide, generated by EEA households' food consumption.

| Share in Food Production Chain (in 2011) | Food Consumption by Households in EEA-Countries | | | | | | | | |
|---|---|---|---|---|---|---|---|---|---|
| | Gross Value Added | Employment | Global Warming Potential | Acidification | Eutrophication | Energy Use | Land Use | Material Use | Water Use |
| Food products | 52% | 80% | 66% | 91% | 94% | 24% | 100% | 79% | 100% |
| Textiles | 0% | 0% | 0% | 0% | 0% | 0% | 0% | 0% | 0% |
| Paper and wood products | 1% | 1% | 0% | 1% | 0% | 1% | 0% | 1% | 0% |
| Energy (related) products | 4% | 1% | 16% | 2% | 1% | 50% | 0% | 5% | 0% |
| Plastics and chemicals | 2% | 1% | 3% | 1% | 1% | 7% | 0% | 3% | 0% |

**Table 4.** *Cont.*

| Share in Food Production Chain (in 2011) | Food Consumption by Households in EEA-Countries | | | | | | | | |
|---|---|---|---|---|---|---|---|---|---|
| | Gross Value Added | Employment | Global Warming Potential | Acidification | Eutrophication | Energy Use | Land Use | Material Use | Water Use |
| Mineral products | 1% | 1% | 1% | 0% | 0% | 1% | 0% | 11% | 0% |
| Metal products | 2% | 1% | 1% | 1% | 0% | 1% | 0% | 2% | 0% |
| Electronics | 1% | 0% | 0% | 0% | 0% | 0% | 0% | 0% | 0% |
| Trade | 10% | 6% | 2% | 0% | 0% | 3% | 0% | 0% | 0% |
| Transport | 11% | 4% | 8% | 3% | 3% | 9% | 0% | 0% | 0% |
| Others | 16% | 5% | 3% | 0% | 0% | 3% | 0% | 0% | 0% |

While the value added in Europe by EEA member country households' food consumption is mainly generated in the food products industry, the value added in Asia/Pacific is generated in the paper and wood products sector as well. Employment outside Europe is located primarily in the food products industry, for which Asia/Pacific and Africa are important regions. In all regions, environmental impacts (e.g., GWP, acidification, land use and water use) are primarily caused by the food production sector, although in Asia/Pacific the share of the energy (related) products sector is equally important as the food production sector.

*5.3. Structural Decomposition Analysis*

The SDA enables quantification of the different contributions of three determinants to indicators of selected environmental footprints and socio-economic effects related to private household consumption in EEA member countries between 1995 and 2011. These determinants are:

- the accumulated intensity (resulting from changes in production technology applied in different industries): it reflects the total environmental or socio-economic unit per unit product used for satisfying the final demand. This accumulated intensity is the sum of the effects resulting from the use of all intermediate inputs at various stages of production along the supply chain, as well as of the effects during final use of each product group (e.g., the amount of $CO_2$ resulting from gasoil production, as well as from gasoil combustion by driving automobiles);
- the consumption structure (or changes in the mix of the product used by private households): it covers the overall effect of changes in the basket of consumed product groups;
- consumption volume (or changes in total volume of products used by private households): it refers to the influence of growth or reduction in total consumption expenditures of private households.

The SDA was carried out applying Equation (7) and using the version product-by-product of the multiregional input–output tables, which describes the intermediate and final use of 200 product groups in the global economy. For this assessment, an aggregated version of the EE-MRIOT was used, in which only the economies of EEA countries as whole and the rest of the world (RoW) are represented. The results of the SDA are not directly comparable with the footprint calculations based on the fully disaggregated EXIOBASE multi-regional input output table (MRIOT). To avoid confusion with absolute footprint values resulting from the fully disaggregated EXIOBASE, we normalized the SDA results. The reference (index 100) is the total footprint of private household consumption in EEA countries in 1995, calculated with the geographically aggregated (EEA+RoW) EXIOBASE.

Absolute changes in footprint levels and absolute contributions of the considered determinants are both calculated from the aggregated EXIOBASE.

### 5.3.1. How to Read the Charts

The solid line shows the sum of the changes caused by these factors, i.e., actual developments of the environmental or socio-economic effects compared to 1995. The bars show changes in the direct and indirect effect "intensities" (accumulated technological effect), changes in the "consumption mix" (structure of total production used by households) and changes in the "consumption volume" (total volume of household expenditure). Each factor has contributed to the change of the assessed environmental or socio-economic macro-indicator compared to 1995 levels. All results are normalized and relate to food-related consumption of private households in EEA member countries.

### 5.3.2. Global Warming Potential (GWP)

Compared to 1995, GHG caused by food-related consumption of households increased slightly in 2000, 2005 and 2007, and were slightly lower in 2010 and 2011 (Figure 4). In 2000, 2005 and 2007, the increase in the volume of food products for household consumption was greater than the increase in GHG emissions from food consumption. In 2010 and 2011, emissions decreased partially even though food consumption expenditure kept increasing. This, however, is not an indication of a decoupling of GWP from the volume of household food expenditure during this period. The SDA shows that the observed decline in GWP is essentially the result of technological innovations in production processes along the global supply chain that led to a reduction in cumulative emission intensities (direct and indirect emissions). Changes in the mix of products used by households also contributed, albeit to a lesser extent and with some intermittency, to the reduction of GWP. In other words, the basket of food-related products defined by private households' food consumption has partly shifted towards groups of goods and services with lower GWP compared to 1995. This shift, although significant, was not constant, with fluctuations over time.

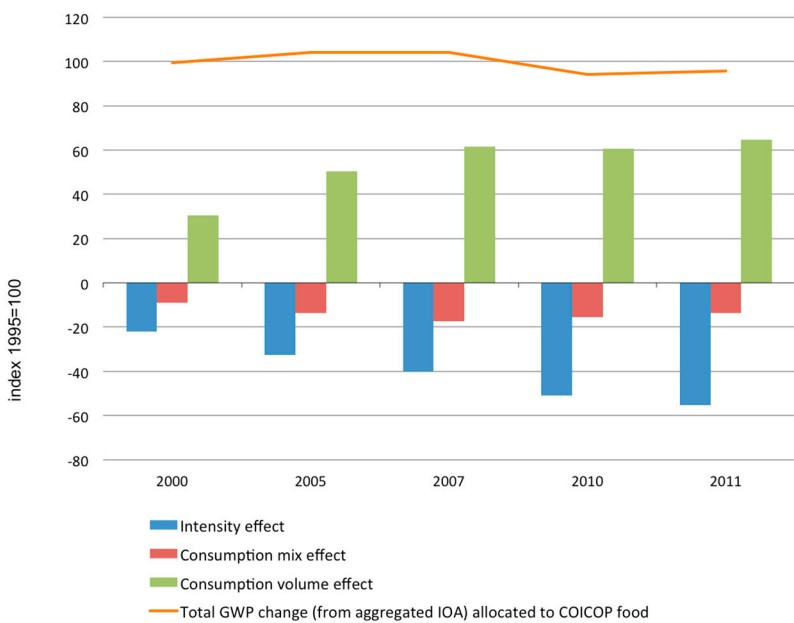

**Figure 4.** Normalized changes in (global) GWP footprint compared to 1995 caused by private households' food-related consumption in the EEA member countries, and decomposition into contributing factors.

In short, although the net total effect shown indicates some decoupling between GWP and food consumption, there are no clear signs of absolute decoupling due to increased total volumes of food consumed.

### 5.3.3. Employment

Compared to 1995, the employment generated worldwide by food consumption of private households increased in 2000 and 2005, before declining to 1995 levels and then slightly increasing between 2010 and 2011 (Figure 5). Increased productivity in global supply chains appears to significantly reduce employment induced by private household food consumption. Changes in the composition of the basket of food-related goods and services also led to declining employment levels, albeit to a lesser extent. Compared to 1995, these observed changes show that food consumption in private households has shifted slightly towards groups of goods and services produced with high labor productivity (i.e., low accumulated labor intensity). However, the increase in the volume of household food consumption expenditure pushed up employment levels, more than offsetting the decline in employment induced by the other two determinants of total change.

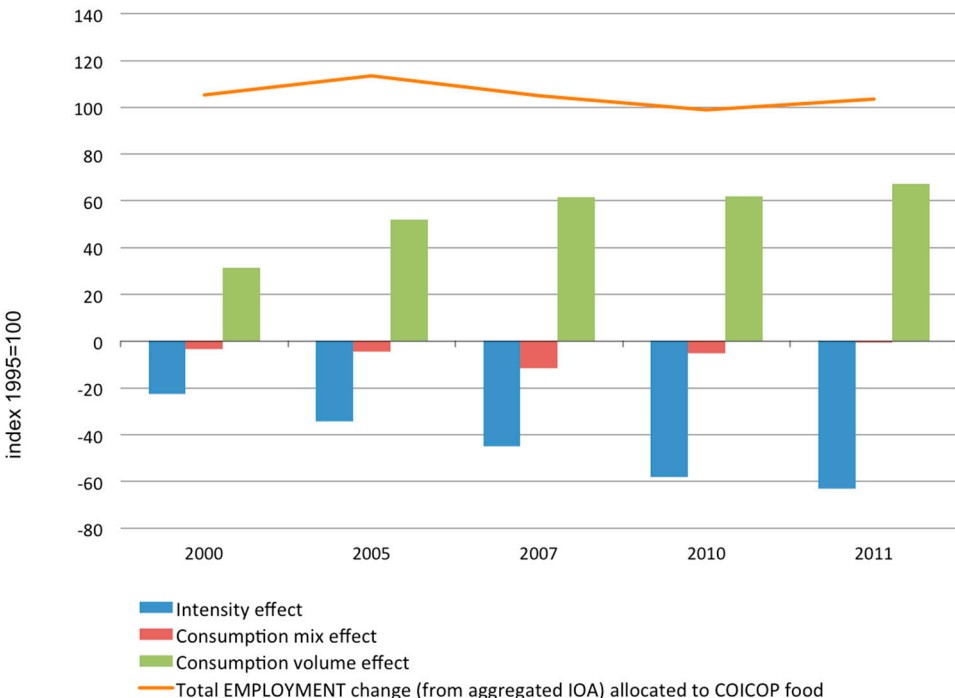

**Figure 5.** Normalized changes in (global) employment footprint compared to 1995 caused by private households' food-related consumption in the EEA member countries, and decomposition into contributing factors.

### 5.3.4. Gross Value Added (GVA)

Compared to 1995, the global GVA generated by food-related consumption of private households steadily increased in the years 2000, 2005, 2007, 2010 and 2011 (Figure 6). The trend is clearer than that for employment, culminating in a total GVA growth of more than 30% compared to 1995. The upward trend in GVA is clearly the result of the increase in the expenditure volume for food products by households which grew more than 50% in 2011 compared to 1995. In contrast, changes in the structure of the products consumed (product mix), as well as in the production technology applied along the global supply chain (cumulative technology effect), had a negative effect on the generation of value added. However, the technology-induced reduction did not go beyond 20%, resulting in a net increase of approximately 30%.

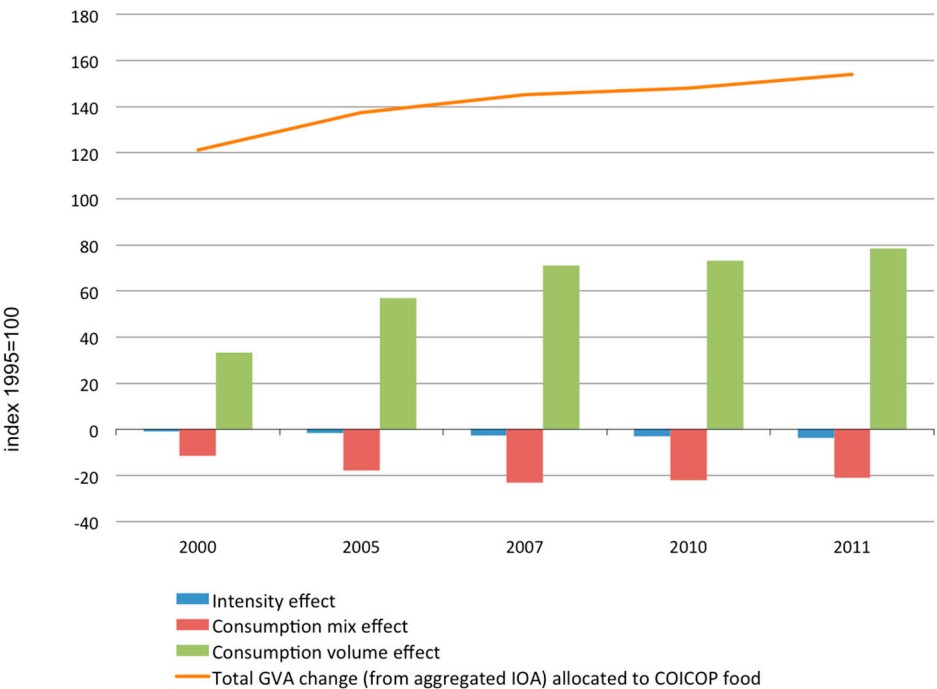

**Figure 6.** Normalized changes in (global) GVA footprint compared to 1995 caused by private households' food-related consumption in the EEA member countries, and decomposition into contributing factors.

In contrast to GHG emissions or employment, changes in production patterns (intensity or direct or indirect effect of the technology) hardly influence the level of GVA induced by food consumption. This might indicate a high degree of efficiency as further improvements did not have considerable economic effects. With regard to the effect induced by changes in the product mix, it can be concluded that food consumption is shifting towards a basket of products and services that has a negative impact on GVA (cheaper food products).

## 6. Discussion and Conclusions

This paper assessed the effects of food consumption in EEA member countries by quantifying a series of socio-economic and environmental indicators (resource use, air emissions, gross value added and employment) and analyzed the main driving forces by means of SDA. The results are based on detailed input–output data from Exiobase, having attributed products to different consumption purposes by means of an ad-hoc allocation. Most studies only briefly discuss the conversion of household consumption expenditure for products supplied by sectors to the COICOP categories. Articles provide neither detailed allocation tables nor supplementary material on the allocation process which limits the reproducibility of results. Since different allocations lead to different results, this can lead to erroneous conclusions about the reliability of assessments and give the impression of a lack of transparency or even arbitrariness of allocations. The missing additional information, the rather sparse discussion of analytical consequences and the missing standardization is a considerable gap in the literature. Therefore, this paper has carefully described the methodology with emphasis on the allocation. As indicated in Section 4.3, assumptions had to be made during construction of the allocation matrix, which, in combination with a lack of detailed data, resulted in some limitations of the analysis. For example, there was a focus on private households rather than a wider range of product groups or services. However, the procedure to develop the allocation matrices is presented in a fully transparent and detailed manner, thus contributing to methodological improvements in integrated assessment, and the application of the different analytical methods answers the main research questions.

## 6.1. Ex-Post Time Series Analysis

The results of the ex-post time series analysis are in line with other studies based on similar types of assessments. These also demonstrate that food consumption is a predominant driver of environmental impacts in categories such as acidification, eutrophication and land use, which are typically consequences of agricultural activities [7,36]. Figure 1 summarizes the assessment of the main socio-economic and environmental indicators of consumption. Animal-based products such as meat, dairy and eggs are the food products which have been identified as responsible for a major part of the impacts (more than 50%) [35]. The analysis in this study identifies the same type of food products as the highest contributors, although not so distinctly and not for all impact categories (Table 2).

With regard to the evolution over time, the growth in consumption in combination with an increasing population have been confirmed as drivers of increased environmental impact in most categories [35]. However, the increase in environmental pressures is overall lower than the increase in consumption which indicates a relative decoupling. Our findings also suggest an absolute decoupling in some impact categories such as land use. A relative decoupling is observed in relation to GWP, which remains more or less stable in absolute terms even though population has increased. Between 1995 and 2017 our assessment suggests even slight tendencies of absolute decoupling. The dynamics of the GWP are mainly influenced by the share of specific types of food products in household expenditures and the GHG emission intensity of the production chain of the respective food products.

## 6.2. Value Chain Analysis

The value chain analysis emphasizes the importance of imports for European household consumption, resulting in environmental pressures being exerted outside of Europe. Our analysis shows, however, that food consumption in EEA member countries generates relatively less environmental pressure abroad than on average, and creates relatively more value added. From the perspective of trade, imports of agricultural and food products in Europe are important contributors to eutrophication, water and land use induced by imports, and these impacts are increasing over time, however less than their value added [35]. There has been growth of emissions in imports between 1995 and 2015 along with growing importance of imports from Asia and related pressures in that region [36]. The latter study observes that 37% of agricultural emissions caused by European consumption patterns occur outside of the EU, which confirms our findings.

## 6.3. Structural Decomposition Analysis

The results of the SDA highlight the environmental and socioeconomic effects resulting from technological development and changing mix of consumed products, and the counteracting increases resulting from expanded consumption.

### 6.3.1. Technology and Mix of Consumed Products (Cumulative Intensity and Effects of the Consumption Mix)

Technological innovation along the global supply chain, as well as changes in the mix of consumed products, has substantially reduced direct and indirect greenhouse gas emissions. However, the effect of the changing consumption mix was intermittent and much less prominent than the cumulative intensity effect resulting from technological innovation in production processes. In other words, since 1995, the basket of food-related products consumed by private households has partly shifted towards goods and services with lower GWP. However, technological progress has been the most important determinant in reducing greenhouse gas emissions. With regard to employment, the results show a pattern similar to the changes observed in terms of GWP. Technological innovation in production processes significantly reduces employment along the supply chain. This effect was supported by changes in the mix of consumed products, however, with a much lower impact. The SDA results show that, in contrast to GHG or employment, the mix of products consumed by households has a

much greater impact on the gross value added induced by the consumption of food products than technological developments. The changes observed for the period between 2000 and 2007 associated with the product mix indicated that household food consumption changed so it is made up of products that induce directly and indirectly a lower value added than in previous periods; however, this trend has reversed since 2007. On the other hand, productivity improvements in production systems have avoided a substantial increase in the costs of direct and indirect intermediate products and services associated with the production of food consumed by households. These trends could be further improved through policy instruments, such as circular economy initiatives which would make it possible to reduce production costs without affecting value added and with a product mix that includes labor-intensive foods with a higher value added (for example, due to the consumption of organic products).

Comparing the SDA results on employment, GVA and GWP highlight that technological progress has resulted in a productivity increase. In combination with changes in consumption patterns, this has increased GVA and reduced GHG emissions as well as employment. In other words, food consumption by households is associated with a production of goods and services with lower production costs, which are significantly cleaner, but at the same time induce less employment.

### 6.3.2. Growth of Consumption

Regardless of the variable, the results of the footprint analysis clearly show that the net effect of food consumption by households is essentially determined by the total volume of consumption of the related product groups. Although there are considerable uncertainties related to the presented SDA, it confirms the role of growth in consumption volumes offsetting reductions in environmental pressures from improved productivity and changes in the product mix. Even though products have become relatively cheaper and less labor-intensive, the increase in consumption to a large extent compensates for the productivity increases, resulting in a more or less constant employment and an absolute growth in gross value added.

### 6.3.3. Conclusions

The analyses presented here contribute to the knowledge base demonstrating that food consumption is an important driver of environmental pressures and impacts. This is the case particularly for impact categories such as acidification, eutrophication and land use, which are related to agricultural activities, with animal-based products like meat, dairy and eggs responsible for a major part of these impacts. However, there has been relative decoupling between environmental pressures and consumption over time, and European food consumption generates relatively less environmental pressures outside of Europe (due to imports) than the average European consumption.

The results of the footprint analysis clearly show that the net effect of food consumption by households is essentially determined by the total volume of consumption of the related product groups. The SDA confirms the role of growth in consumption volumes offsetting reductions in environmental pressures from improved productivity and changes in the product mix.

These analyses also highlight trade-offs rather than synergies between employment, environment and economic growth, especially between economic growth and the environment in a "full world" [37,38]. This is in line with recent findings of the International Resource Panel who, in the latest resource outlook, analyzed drivers of the material footprint in seven world regions and globally [39]. For two periods (1990–2000 and 2000–2016), economic growth ("affluence") was by far the most prominent driver of domestic extraction of natural resources in Europe, offsetting considerable technological gains.

Even though we acknowledge that our findings are rather pessimistic in relation to mainstream discussions about the potentials of a European green economy, the empirical pattern seems to be clear and in line with similar assessments, e.g., [36,39,40]. This highlights the need to increase research on IO data and models and highly detailed analytical approaches that enable analysis of direct and indirect

impacts of and interlinkages within the food system. These approaches and models can have an important role within the new EU research and innovation agenda associated with the EU Farm to Fork Strategy and Horizon Europe 2021–2027 (Cluster 6: Food, bio-economy, natural resources, agriculture and environment). The Farm to Fork Strategy (COM(2020)381) adopts an objective of achieving a neutral or positive environmental impact food chain, and Horizon Europe's Mission Board for 'Soil health and food' indicates a "20–40% reduced global footprint of EU's food and timber imports on land degradation" as a 'mission' for EU research and innovation [5]. Such tools used in combination with behavioral and social research can provide an advanced knowledge base to support these ambitious policy and research objectives.

**Author Contributions:** All authors contributed equally to the conceptualization, methodology, formal analysis, data curation, visualization, draft preparation, review and editing. All authors have read and agreed to the published version of the manuscript.

**Funding:** This research received no external funding and was undertaken as part of the joint work programme of the European Environment Agency and European Topic Centre on Waste and Materials in a Green Economy.

**Acknowledgments:** The authors would like thank colleagues at the European Environment Agency, Joint Research Centre and Eurostat for useful discussions. Usual disclaimer apply.

**Conflicts of Interest:** The authors declare no conflict of interest.

## Appendix A. Direct and Indirect Pressures per Euro Expenditure within Different Household Consumption Categories

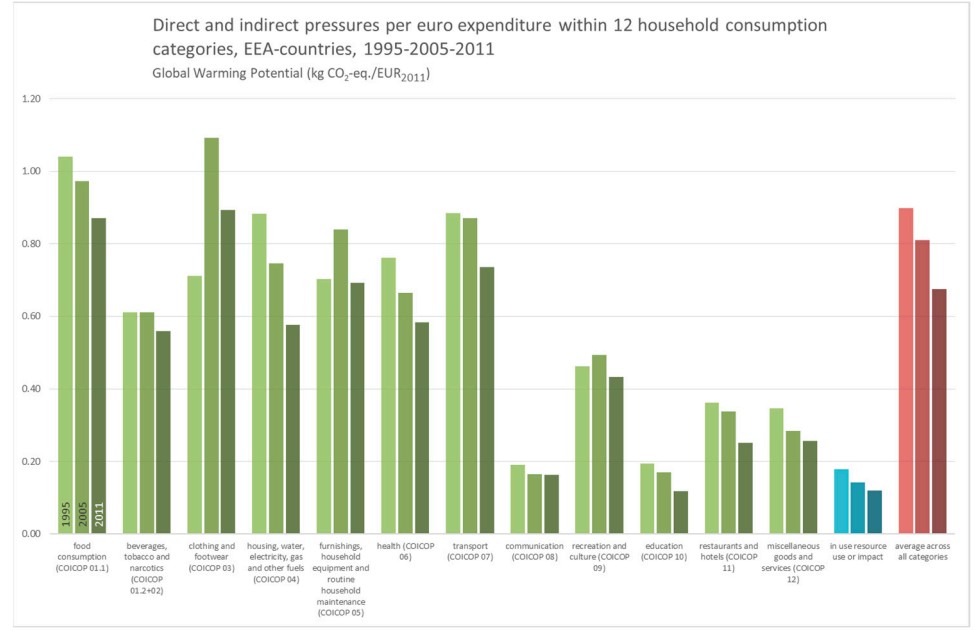

**Figure A1.** Direct and indirect pressures (global warming potential) per euro expenditure within 12 household consumption categories, EEA countries, 1995–2005–2011.

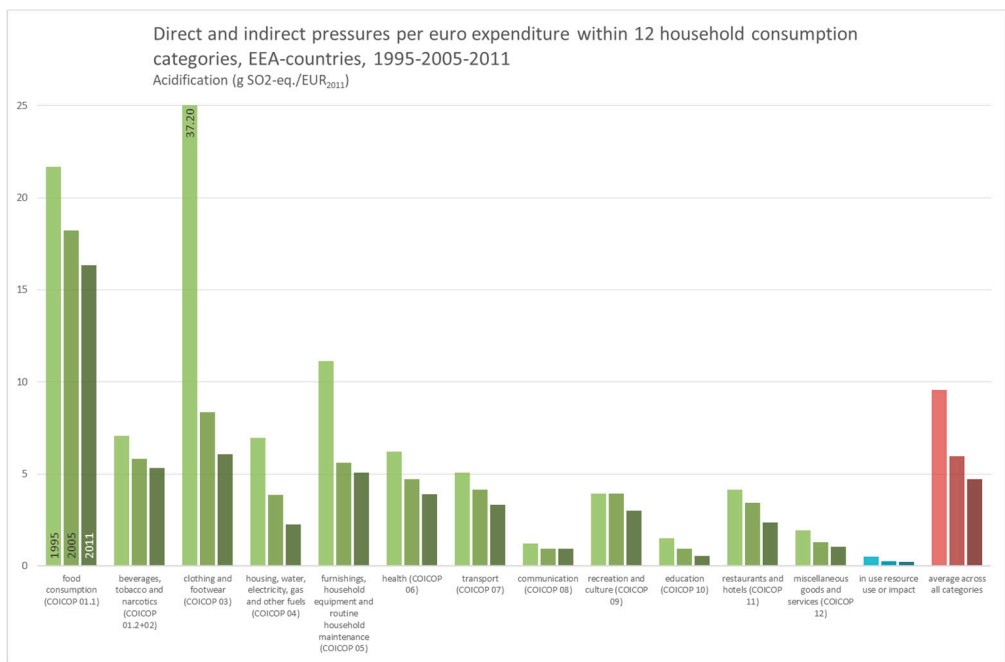

**Figure A2.** Direct and indirect pressures (acidification) per euro expenditure within 12 household consumption categories, EEA countries, 1995–2005–2011.

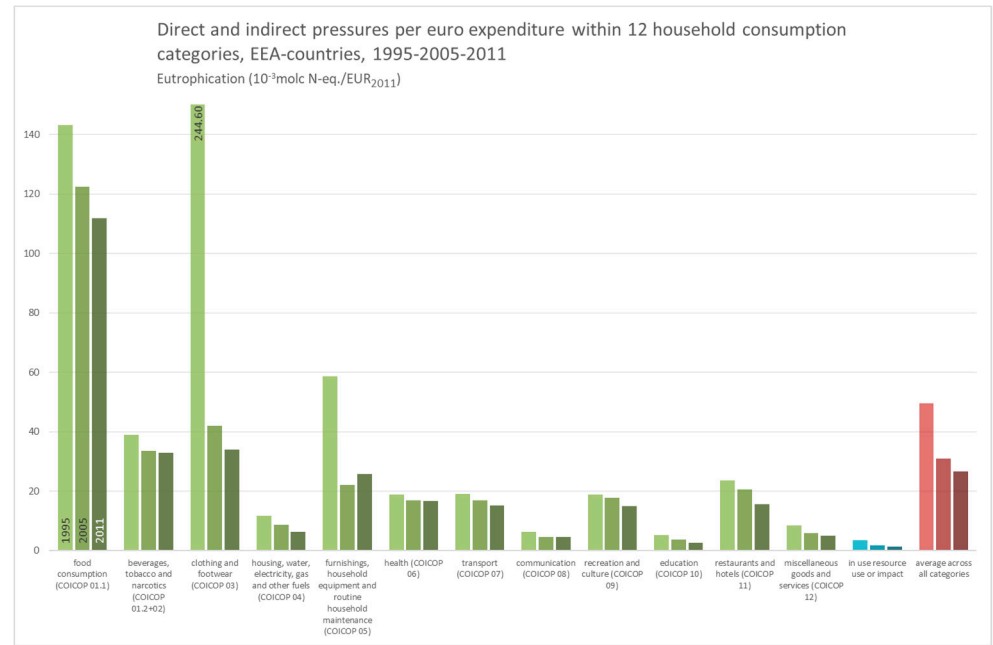

**Figure A3.** Direct and indirect pressures (eutrophication) per euro expenditure within 12 household consumption categories, EEA countries, 1995–2005–2011.

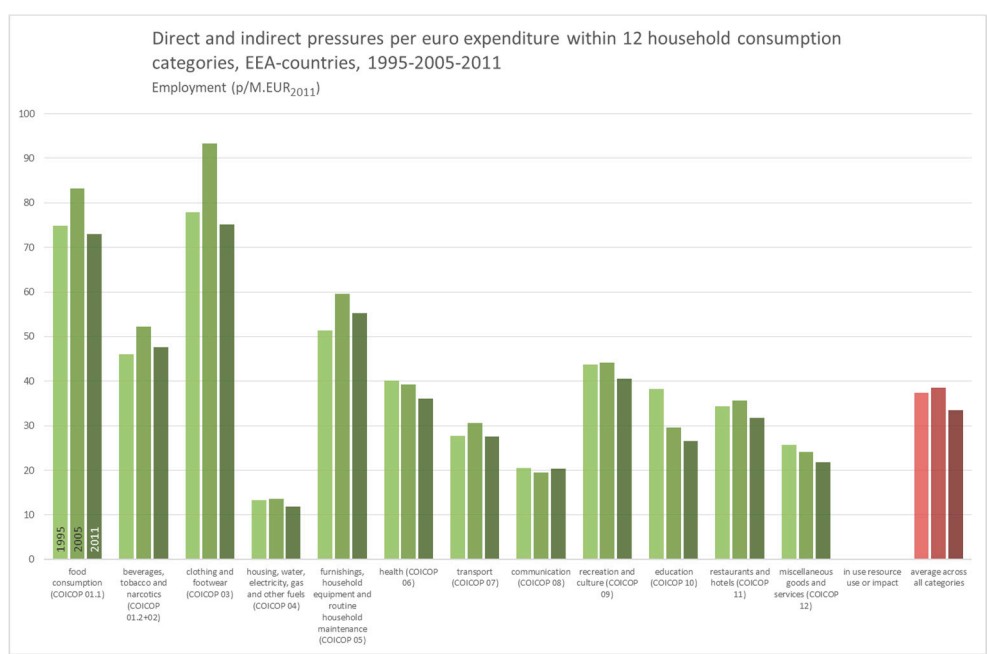

**Figure A4.** Direct and indirect pressures (employment) per euro expenditure within 12 household consumption categories, EEA countries, 1995–2005–2011.

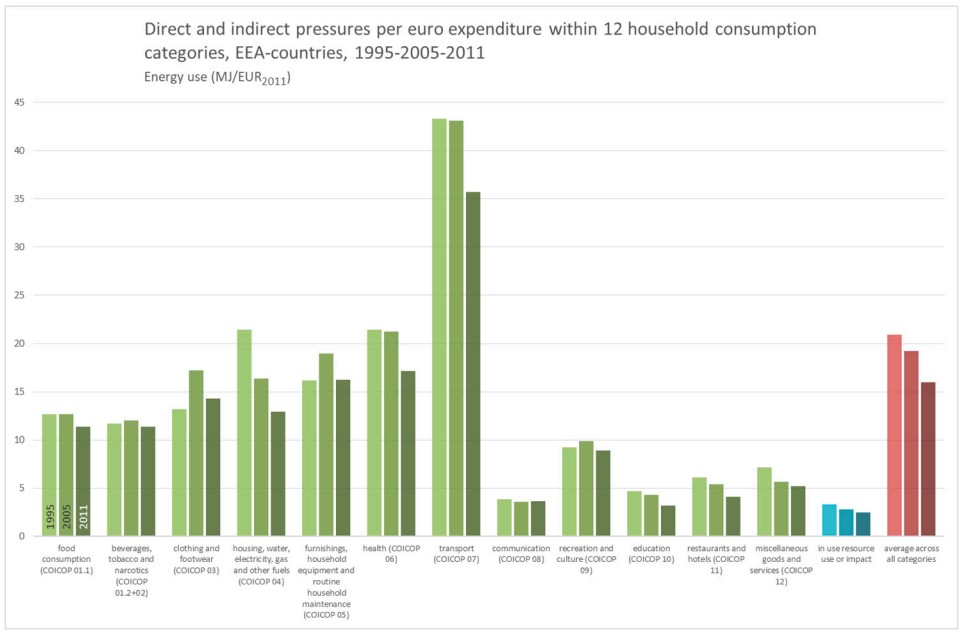

**Figure A5.** Direct and indirect pressures (energy use) per euro expenditure within 12 household consumption categories, EEA countries, 1995–2005–2011.

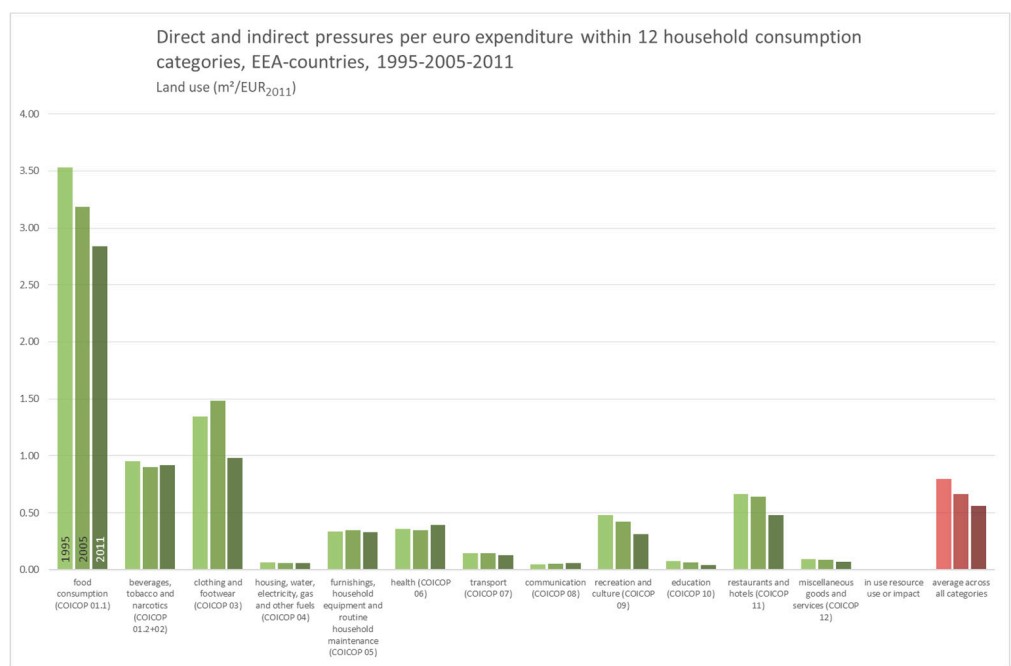

**Figure A6.** Direct and indirect pressures (land use) per euro expenditure within 12 household consumption categories, EEA countries, 1995–2005–2011.

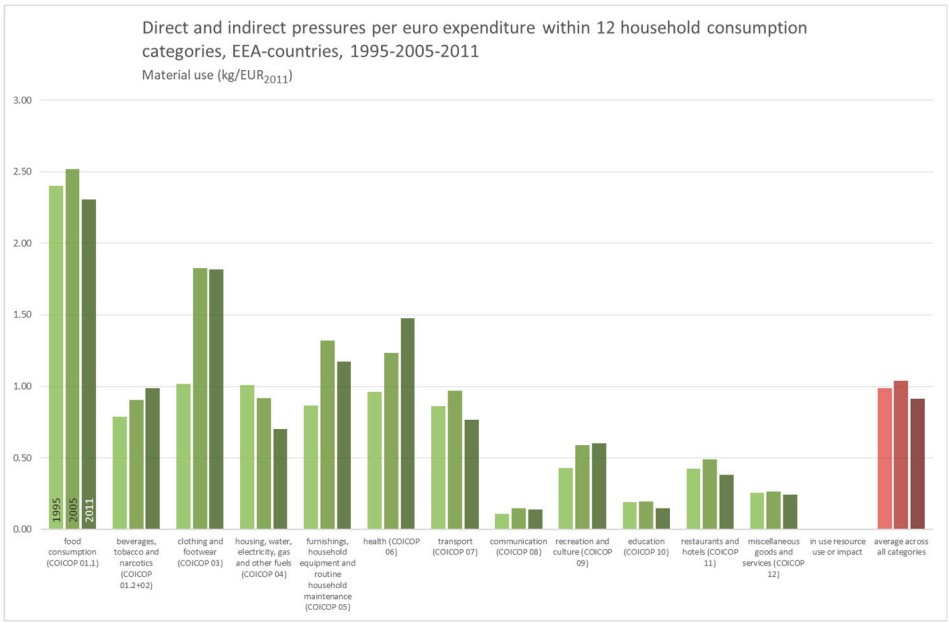

**Figure A7.** Direct and indirect pressures (material use) per euro expenditure within 12 household consumption categories, EEA countries, 1995–2005–2011.

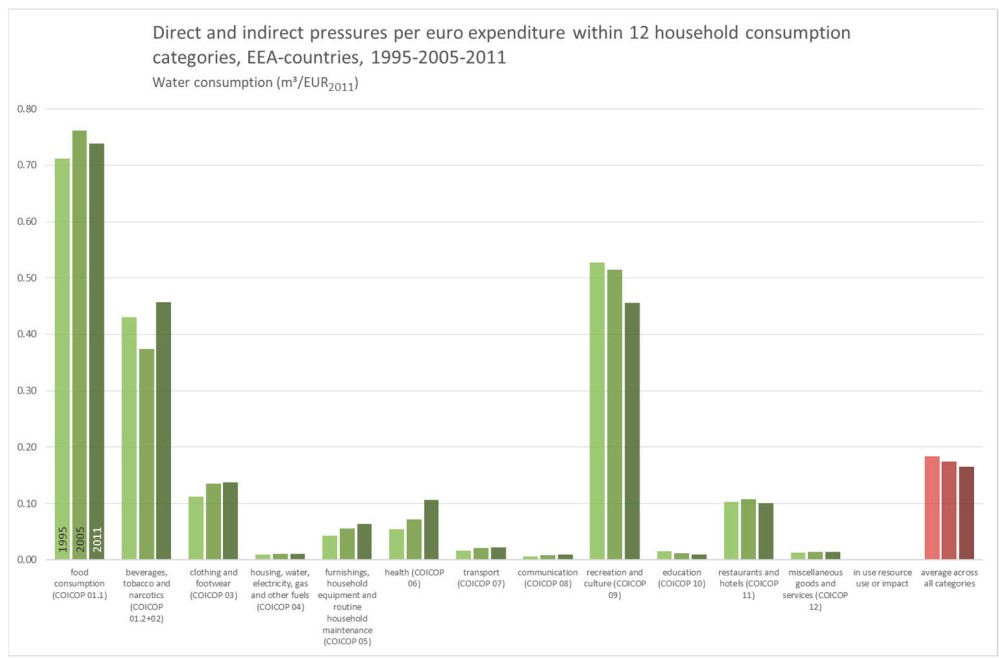

**Figure A8.** Direct and indirect pressures (water consumption) per euro expenditure within 12 household consumption categories, EEA countries, 1995–2005–2011.

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
