# Peer review of "Driving Forces of Changing Environmental Pressures from Consumption in the European Food System"

_sustainability, doi:10.3390/su12198265_

Round 1

Reviewer 1 Report

Dear Authors,

thank you very much for the interesting paper and the wide-ranging analysis. The most crucial point in your analysis refers to extrapolation until 2017 (see below) – if it is possible to be consistent throughout the paper, this would improve applicability of results significantly.

After reading the introduction, I wondered if you are really the first ones who “identifies the driving forces of observed changes in environmental and socio-economic effects”. This point refers to the innovative potential of the manuscript. Probably you could clarify this point already in the introduction, what is the innovation in your analysis compared to existing approaches and publications?

Chapter 2, State of the art is good. You briefly describe approaches “to analyse the environmental impacts of human food consumption”

Line 240: Chapter 3, assessment questions: In my opinion the term “assessment question” is misleading here, these are rather your research questions. To answer them, you use several methods and, of course, you assess input-output relations; if you agree, you should change this in the whole document, e.g., line 329: “Since the assessment questions focused on the effects …”

Line 301: Definition food system: Considering the headline of the chapter, I would expect at least a brief general definition of the term “food system”, you are providing immediately – as you write – a very specific “definition” of  a “food system in terms of COICOP categories”. This is rather a categorization than a definition.

Line 424-426: You mention several assumptions for your approximations (based on Miller, R., Blair, P. R. (2009)?), you could include sources from literature forming the basis for these assumptions (used in comparable studies).

Line 460-466: Extrapolation, linear trend: The results of the trend extrapolation could be included (in the Appendix, model fit/coefficient of determination, confidence intervals etc.).

Line 543: I was really surprised that you indexed your result to 2017 – I would have expected 100 equals the first year of your data set, this would be the usual approach. Please change that – in this case you could also include confidence intervals (e.g., 95%) as you extrapolated your data for 2017 and I guess the results out of the linear trend might be connected with a certain bandwidth (if I understood your analysis correctly; line 460, see above).

Line 575: To show environmental pressure, you are not extrapolating until 2017 – why not? Is it not possible? From our present perspective, changes between 1995 and 2011 are interesting, but much less interesting compared to estimations reaching into the present. I think this is the most critical point in your manuscript – I hope it is possible to improve that and make all results comparable.

In footnote no. 4 & 5 you write: “The discussion on … is limited to the period 1995-2011, as the 2017-intensity is an extrapolation result of this time period (see Section 4.5).” In this section you should present the results, the discussion about them should be found in the discussion chapter. Nevertheless, you could provide extrapolations for 2017 also in the result section (see above)?

Line 598…: Chapter 5.2. Value Chain Analysis: Confirming Literature a value chain represents the entire input-output processes that brings a product or service from initial conception to the consumer. From my understanding, a value chain analysis would therefore focus on the stages within the value chain (where value is created: agriculture, food production, trade, logistics, etc.), but you write: “A value chain analysis identifies the part of the footprint that is located outside EEA member …”. This could be one part of the VCA, but it shouldn’t a VCA mainly focus on the creation of value within a specific production to consumption chain? Probably you could something to your headline or change it, to make the point clearer that you are not conducting a complete VCA.

Line 655, SDA: See comments about extrapolation of number to present situation above.

General comment: It is hard to connect your theoretical explanations of what you will do (e.g., 4.4. Extended input-output-analysis, line 413) with your results, as there are neither references to the number of formulas you present nor to the used abbreviations in the results’ section. E.g., refers “intensity of environmental pressures” in Line 575 to e^acc or only to parts of e^acc in Formula (5) or is it connected to Formula (7)?

I really appreciated your discussion chapter, it is however an aggregate of discussion (in comparison with existing findings) and conclusions. Probably, you should name the chapter “Discussion and conclusions”. You should also add a chapter about limitations (you mentioned some throughout the text).

Line 751: You write: “and identified the main driving forces by means of structural decomposition analysis”. As driving forces are already mentioned in the title of your manuscript, you should explicitly name them and explain, why these are the MAIN driving forces.

Figures in Appendix should be named Figure A1, A2 … (also reference in text). Couldn’t the figures 1 to 1 H, also be integrated into one single table?

Language: Seems to be fine, but you should use exclusively BE or AE (e.g.: “jeopardize” in Line 32, “behaviour” throughout the manuscript).

Author Response

Dear reviewer,

Thank you for the review of our manuscript and the valuable comments and suggestions for the improvement of our draft article.

The team of nine authors from four different countries has made an effort to carefully address each point raised in your review. Please find below a table in which we have listed each item and the corresponding reply.

Item

Reply

The most crucial point in your analysis refers to extrapolation until 2017 (see below) – if it is possible to be consistent throughout the paper, this would improve applicability of results significantly.

We acknowledge the fact that extrapolation until 2017 is not consistent throughout the paper. The extrapolation of the coefficients (intensities) from 2011 to 2017 was required due to lack of data. The problem is that no IO-data are available (thus no data for intensities) after 2011. As such, for intensities results a detailed analysis until 2017 has no added value as we would only analyse our own extrapolations. We have elaborated the explanation in par. 4.5 to clarify this.

After reading the introduction, I wondered if you are really the first ones who “identifies the driving forces of observed changes in environmental and socio-economic effects”. This point refers to the innovative potential of the manuscript. Probably you could clarify this point already in the introduction, what is the innovation in your analysis compared to existing approaches and publications?

We clarified in the induction that a major contribution of the paper is a very detailed, analytical and complete description of the methodology, and in particular of the process leading to the allocation matrices, the latter being the basic tool for the assessment. This is an added value of the paper with respect to other works on integrated assessment using EE-IO approaches, which are less detailed and transparent. On the side of the driving forces analysis, it should now be more clear that we are not the first using Structural Decomposition analysis. This is also emphasised in the description of the SDA in section 5.3.

Chapter 2, State of the art is good. You briefly describe approaches “to analyse the environmental impacts of human food consumption”

Line 240: Chapter 3, assessment questions: In my opinion the term “assessment question” is misleading here, these are rather your research questions. To answer them, you use several methods and, of course, you assess input-output relations; if you agree, you should change this in the whole document, e.g., line 329: “Since the assessment questions focused on the effects …”

The text has been revised to address the comment and the term research questions used where appropriate.

Line 301: Definition food system: Considering the headline of the chapter, I would expect at least a brief general definition of the term “food system”, you are providing immediately – as you write – a very specific “definition” of  a “food system in terms of COICOP categories”. This is rather a categorisation than a definition.

A definition of the food system has been added and the text amended in line with the comment.

Line 424-426: You mention several assumptions for your approximations (based on Miller, R., Blair, P. R. (2009)?), you could include sources from literature forming the basis for these assumptions (used in comparable studies).

The assumptions mentioned in lines 424-426 are the standard assumptions of the Leontief input-output model. In addition to Miller and Blair (2009), we now also refer to Oosterhaven (1996) who discusses more specifically these general assumptions and their implications.

Line 460-466: Extrapolation, linear trend: The results of the trend extrapolation could be included (in the Appendix, model fit/coefficient of determination, confidence intervals etc.).

We refer to our reply on the first comment. In this specific case, analysing in detail the extrapolated results until 2017 would not add value.

Line 543: I was really surprised that you indexed your result to 2017 – I would have expected 100 equals the first year of your data set, this would be the usual approach. Please change that – in this case you could also include confidence intervals (e.g., 95%) as you extrapolated your data for 2017 and I guess the results out of the linear trend might be connected with a certain bandwidth (if I understood your analysis correctly; line 460, see above).

Correct. We have changed the graph accordingly. It is not possible to add the confidence intervals to the graph, but we have included a clarification about this in par. 4.5.

Line 575: To show environmental pressure, you are not extrapolating until 2017 – why not? Is it not possible? From our present perspective, changes between 1995 and 2011 are interesting, but much less interesting compared to estimations reaching into the present. I think this is the most critical point in your manuscript – I hope it is possible to improve that and make all results comparable.

We refer to the clarification added to par. 4.5. We analyse the environmental intensities only until 2011 because no data are available for later years. The environmental footprints are analysed until 2017, using a combination of expenditure data and extrapolated intensities.

In footnote no. 4 & 5 you write: “The discussion on … is limited to the period 1995-2011, as the 2017-intensity is an extrapolation result of this time period (see Section 4.5).” In this section you should present the results, the discussion about them should be found in the discussion chapter. Nevertheless, you could provide extrapolations for 2017 also in the result section (see above)?

We refer to the previous response w.r.t. this issue.

Line 598…: Chapter 5.2. Value Chain Analysis: Confirming Literature a value chain represents the entire input-output processes that brings a product or service from initial conception to the consumer. From my understanding, a value chain analysis would therefore focus on the stages within the value chain (where value is created: agriculture, food production, trade, logistics, etc.), but you write: “A value chain analysis identifies the part of the footprint that is located outside EEA member …”. This could be one part of the VCA, but it shouldn’t a VCA mainly focus on the creation of value within a specific production to consumption chain? Probably you could something to your headline or change it, to make the point clearer that you are not conducting a complete VCA.

That is partly correct. A VCA assesses the whole value chain of a product or service. But the focus of the VCA can be different: it can indeed focus on the creation of value, but it is also used to identify the phases where the major environmental impact occurs, or it can distinguish between value and/or environmental impact occurring in and outside a specific region. We have used the VCA approach for the latter. We acknowledge the fact that the first sentence can be interpreted, and have slightly reformulated this. To be clear, we have conducted a complete VCA, but focussed the analysis on the geographical scope.

Line 655, SDA: See comments about extrapolation of number to present situation above.

on extrapolation see reply above

General comment: It is hard to connect your theoretical explanations of what you will do (e.g., 4.4. Extended input-output-analysis, line 413) with your results, as there are neither references to the number of formulas you present nor to the used abbreviations in the results’ section. E.g., refers “intensity of environmental pressures” in Line 575 to e^acc or only to parts of e^acc in Formula (5) or is it connected to Formula (7)?

The numbers of the equations presented in sections 4.4 and 4.6 have been inserted in different parts of chapter 5 (sections 5.1 to 5.3). Some minor edits have also been made to the text.

I really appreciated your discussion chapter, it is however an aggregate of discussion (in comparison with existing findings) and conclusions. Probably, you should name the chapter “Discussion and conclusions”. You should also add a chapter about limitations (you mentioned some throughout the text).

The chapter has been renamed and restructured to address the comment. There is now a sub-section called conclusions. We did not add an additional sub-section on limitations as summarising them in an additional section would just repeat text and add to the length of the manuscript. Instead we ensured that the limitations have been summarised and presented more explicitly in the opening paragraph of the chapter.

Line 751: You write: “and identified the main driving forces by means of structural decomposition analysis”. As driving forces are already mentioned in the title of your manuscript, you should explicitly name them and explain, why these are the MAIN driving forces.

The text has been revised to address the comment. Throughout we now state that we analysed the main driving forces and specified them. The justification for the selection of these driving forces is explained in section 4.6 as they related to the determinants of change analysed in the SDA.

Figures in Appendix should be named Figure A1, A2 … (also reference in text). Couldn’t the figures 1 to 1 H, also be integrated into one single table?

The figures are renamed according to the instructions for authors. Integrating them in one single table is not beneficial for the readability and drawing of conclusions.

Language: Seems to be fine, but you should use exclusively BE or AE (e.g.: “jeopardize” in Line 32, “behaviour” throughout the manuscript).

The language has been checked and spelling is now consistent (UK English)

I hope our responses and the modifications in the manuscript meet your expectations. 

Thank your for your time and efforts.

On behalf of the team of authors,

Yours sincerely

philipp schepelmann

Reviewer 2 Report

The manuscript entitled “Driving forces of changing environmental pressures from consumption in the European food system” presents an interesting issue, and may be interesting for readers, however it requires some amendments. The manuscript is well written but it is a little too long.

ABSTRACT:

  • The most important findings (results) should be more emphasized.

INTRODUCTION:

  • The scientific gap should be emphasized. The novelty of the study should be indicated.
  • Despite of the fact, that EU28 plus Norway, Iceland, Switzerland, Lichtenstein and Turkey are the member countries of EEA, the proper justification of analyses the data from Turkey and Iceland (and other countries) must be presented. Especially in case of these two countries it seems to be odd to analyses all this countries together. The socioeconomic background of these countries are different so it could interfere the results.

RESULTS and DISCUSSION:

  • The most prominent results are presented in table 2 – this table should be more emphasized
  • Figure 2 and figure 3 should be presented in a table (if possible)
  • In discussion section there are some parts without any references support. The proper references must be added.

CONCLUSIONS:

  • Taking into account the length of the manuscript, the sub-chapter entitled “conclusion” will be expected

Other comments:

  • Authors should follow the Instructions for authors while preparing their manuscript (including way of citation – please see lines 75, 81).
  • Authors should follow the Instructions for authors while preparing their references list
  • The figures are too small to read (in annex)
  • Please remove the title form figures (the title below figure is sufficient)
  • The tables should be prepared according to the Instructions for authors

Author Response

We are very grateful to the reviewer for the very useful comments and suggestions. The amendments we have made in response to these comments and suggestions are described in detail point by point in the following table:

The manuscript entitled “Driving forces of changing environmental pressures from consumption in the European food system” presents an interesting issue, and may be interesting for readers, however it requires some amendments.

The manuscript is well written but it is a little too long.

On the length of the paper, we deemed it necessary to provide the reader with a very detailed and complete description of the methodology and in particular the process leading to the allocation matrices. This is an added value of the paper with respect to other works on integrated assessment using EE-IO approaches. The results and the discussion are presented in a relatively concise way with respect to the rich set of results. As a consequence, an attempt to shorten the paper might result in a loss of relevant parts or lower clarity. We have revised the text to be more concise where possible.

ABSTRACT: The most important findings (results) should be more emphasised.

We have changed the abstract to better highlight the main findings.

INTRODUCTION: The scientific gap should be emphasised. The novelty of the study should be indicated.

We have now highlighted in the introduction the novelty and originality of the work in particular in its methodological and analytical components. We have also highlighted major results.

Despite of the fact, that EU28 plus Norway, Iceland, Switzerland, Lichtenstein and Turkey are the member countries of EEA, the proper justification of analyses the data from Turkey and Iceland (and other countries) must be presented. Especially in case of these two countries it seems to be odd to analyses all this countries together. The socioeconomic background of these countries are different so it could interfere the results.

The geographical scope was selected to ensure that the analysis covered the largest share of EEA countries' economies as possible. In terms of illustrating the method and the key results and conclusions, the inclusion of non-EU countries has not significantly affected the results as they align well with other EU28 estimates. This justification has been added to the relevant footnote.

RESULTS and DISCUSSION:

The most prominent results are presented in table 2 – this table should be more emphasised

Table 2 is now better referenced within the text.

Figure 2 and figure 3 should be presented in a table (if possible)

Presenting these results in table format is indeed possible, however we feel some important insights (e.g. evolution over time, differences between the environmental indicators and link with GVA) are more clear in a graphical format.

In discussion section there are some parts without any references support. The proper references must be added.

The discussion section has been revised to a discussion and conclusions section and referencing checked and amended where needed. This makes a clear distinction between the conclusions drawn from this research (where referencing is not required) and comparison of the results to other studies (where complete referencing is required)

CONCLUSIONS:

Taking into account the length of the manuscript, the sub-chapter entitled “conclusion” will be expected

Section 6 has been restructured to a discussion and conclusions section.

Other comments:

Authors should follow the Instructions for authors while preparing their manuscript (including way of citation – please see lines 75, 81).

The entire paper was revised following the instruction for authors

Authors should follow the Instructions for authors while preparing their references list

References list was updated and revised

The figures are too small to read (in annex)

Indeed, a printed version of the graphs is too small for reading. Technically it is possible to read the text of the annexed figures by enlarging the graphs on a monitor. Therefore, we will ask the editors, whether supplementing the graphs as separate files would be more appropriate.

Please remove the title form figures (the title below figure is sufficient)

Done for the SDA figures. The Excel files (Supp Mat) containing all data used to build the SDA graphs and the graphs themselves have been updated, i.e. the graphs without title can be found there too.

The tables should be prepared according to the Instructions for authors

We have updated all tables according to the Instructions for authors.

We hope our responses and modifications to the manuscript meet your expectations.

Thank you for your time and efforts.

On behalf of the team of authors,

Yours sincerely,

philipp schepelmann

Round 2

Reviewer 1 Report

Dear Authors,

thank you for your revision and detailed reply. Even though you couldn't solve all points I mentioned in my last review report, I see now why you had to conduct your analysis as you did.

Kind regards